

# Statistics on clouds and their relation to thermodynamic conditions at Ny-Ålesund using ground-based sensor synergy

Tatiana Nomokonova[1], Kerstin Ebell[1], Ulrich Löhnert[1], Marion Maturilli[2], Christoph Ritter[2], and Ewan O'Connor[3]

[1]Institute for Geophysics and Meteorology, University of Cologne, Cologne, Germany
[2]Alfred Wegener Institute Helmholtz Centre for Polar and Marine Research, Potsdam, Germany
[3]Finnish Meteorological Institute, Helsinki, Finland, and Meteorology Department, University of Reading, Reading, United Kingdom

**Correspondence:** Tatiana Nomokonova (tnomokon@uni-koeln.de)

**Abstract.** The French-German Arctic Research Base AWIPEV at Ny-Ålesund, Svalbard, is an unique station for monitoring cloud related processes in the Arctic. For the first time, data from a set of ground-based instruments at AWIPEV observatory are analyzed to characterize the vertical structure of clouds. For this study, a 14-month dataset from Cloudnet combining observations from a ceilometer, a 94 GHz cloud radar and a microwave radiometer, is used. The total cloud occurrence of
∼81 %, with 44.8 % of multi-layer and 36 % of single-layer clouds was found. Among single-layer clouds the occurrence of liquid, ice and mixed-phase clouds was 6.4 %, 9 % and 20.6 %, respectively. It was found, that more than 90 % of single-layer liquid and mixed-phase clouds have LWP values lower than 100 and 200 g m$^{-2}$, respectively. Mean values of IWP for ice and mixed-phase clouds were found to be 273 and 164 g m$^{-2}$, respectively. The different types of single-layer clouds are also related to in-cloud temperature and relative humidity under which they occur. Statistics based on observations are
compared to the ICON model output. Distinct differences in liquid phase occurrence in observations and the model at different environmental temperatures leading to higher occurrence of pure ice clouds and lower occurrence of mixed-phase clouds in the model at temperatures between -20° and -5°C become evident. The analyzed dataset is useful for satellite validation and model evaluation.

## 1 Introduction

Clouds play a crucial role in the energy and in the hydrological cycle. On the one hand, clouds scatter solar radiation back to space leading to a shortwave cooling effect at the surface. On the other hand, clouds conserve longwave radiation and therefore warm the surface in the longwave. The impact of clouds on the energy budget depends on their macrophysical (cloud thickness, cloud top and cloud base altitudes) and microphysical (phase, size, and concentration) properties (Sedlar et al., 2012; Dong et al., 2010).

One of the most important cloud characteristics defining the radiative properties is cloud phase composition (Sun and Shine, 1994; Yoshida and Asano, 2005). In general, liquid containing clouds exhibit a stronger cloud radiative effect than ice clouds (Shupe and Intrieri, 2004). The phase partitioning is especially essential in the Arctic region, where liquid and mixed-phase



clouds can persist for several days (Morrison et al., 2012). Shupe and Intrieri (2004) and Intrieri et al. (2002) showed that during the polar winter liquid-containing clouds significantly influence the net cloud radiative effect and lead to an enhanced warming near the surface. The authors also reported that in midsummer the cloud-driven shortwave radiation cooling dominates over longwave warming.

The net cloud radiative forcing in the Arctic influences sea ice coverage and leads to more open water that in turn affects a heat exchange between ocean and atmosphere (Serreze et al., 2009; Kapsch et al., 2013). Extended periods of open ocean increase the moisture content in the atmosphere and therefore, might enhance cloud coverage (Rinke et al., 2013; Palm et al., 2010; Kay and Gettelman, 2009; Mioche et al., 2015; Bennartz et al., 2013).

Beyond the radiative feedbacks clouds are crucial for precipitation formation that significantly affects the Arctic climate.
Precipitated water forms rivers and sustains a glacier flow into the sea, and thus influences the salinity of the Arctic ocean. Being essential for snowmelt (Zhang et al., 1996), sea-ice reduction (Kay et al., 2008; Kay and Gettelman, 2009) and affecting the permafrost stability, Arctic clouds have a significant impact on productivity and variety in marine and terrestrial environments and thus influence the Arctic ecosystem (Vihma et al., 2016).

Formation of Arctic clouds is a complicated process associated with aerosol-cloud interactions, turbulence, phase transitions,
heat and moisture exchanges between the surface and the atmosphere (Morrison et al., 2012). The interaction of clouds with radiation and aerosols remains the largest uncertainties in radiative forcing models (Myhre et al., 2013; Walsh et al., 2009). Many of the processes are not well resolved in global climate models (Vihma et al., 2016; Klein et al., 2009) indicating that parameterization of cloud properties still needs improvements (Morrison et al., 2008; Shupe et al., 2011).

Better understanding of Arctic cloud processes and feedbacks requires long-term and accurate observations (Devasthale
et al., 2016; Blanchard et al., 2014). In particular, the knowledge of the vertical cloud structure and phase is crucial for an estimation of the cloud radiative impact (Turner et al., 2018; Liu et al., 2012; Shupe and Intrieri, 2004; Curry et al., 1996). In order to retrieve information on the vertical distribution of clouds and their properties active remote sensing instruments such as lidars and cloud radars have to be exploited (Protat et al., 2006). Using ground- and ship-based remote sensing measurements, Shupe et al. (2011), Shupe (2011) and Intrieri et al. (2002) have provided statistics on cloud phase, cloud macrophysical and
microphysical properties for several Arctic sites and the Beaufort sea within the SHEBA (Surface Heat Budget of the Arctic Ocean) program (Uttal et al., 2002).

Several studies based on satellite observations analyzed Arctic cloud properties including cloud phase. Liu et al. (2012) and Mioche et al. (2015) characterized the vertical and seasonal variability of Arctic clouds using the CloudSat 94 GHz radar and the Cloud-Aerosol Lidar and Infrared Pathfinder (CALIPSO). Liu et al. (2017) combined active space-borne and ground-based
measurements to compare annual cycles of vertical distribution of cloud properties at the Alaskan site Barrow and the Canadian site Eureka. Blanchard et al. (2014) combined the both sets of observations at Eureka station in the high Arctic and showed the seasonal variability of the vertical distribution of clouds and monthly cloud fraction.

Despite of high value of the satellite cloud observations, it is often difficult to capture low-level clouds, which frequently occur in the Arctic (Shupe et al., 2011), with space-borne instruments (Mioche et al., 2015). Ground-based remote sensing
observations can provide more detailed information here. However, there are only a few Arctic sites that provide long-term,



continuous information about the vertical cloud structure using the combination of ground-based remote sensing measurements. Such sites are located, for example, in Barrow (Alaska; Verlinde et al., 2016; de Boer et al., 2009), in Atqasuk (Alaska; Doran et al., 2006), in Eureka (Canada; de Boer et al., 2009), and in Summit (Greenland; Shupe et al., 2013). Shupe et al. (2011), for instance, compared the occurrence and cloud macrophysical properties of six observatories in the Arctic.

One of the Arctic cloud observation sites is based in Ny-Ålesund (78.92°N, 11.92°E), which is located on the island of Spitsbergen in Svalbard, Norway and comprises several international research stations. Ny-Ålesund is situated at the coastline of Svalbard close to a fjord, ocean, mountains, and thus, its climate is significantly influenced by diabatic heating from the warm ocean (Serreze et al., 2011; Mioche et al., 2015) and by the surrounding orography (Maturilli and Kayser, 2016a). Maturilli and Kayser (2016a) have already shown a highly pronounced warming and moistening of the tropospheric column in the Svalbard

region. Analyzing a 22-year dataset (1993-2014) from radiosondes the authors found that during winter time there has been a significant increase of atmospheric temperature (up to 3 K per decade) and mean integrated water vapor (+0.83±1.22 kg m$^{-2}$ per decade).

Shupe et al. (2011) analyzed cloud statistics at Ny-Ålesund based on data from a micropulse lidar only. Recently, Yeo et al. (2015) have investigated the relation between cloud fraction and surface long-wave and shortwave radiation fluxes at Ny-

Ålesund using data from the lidar. Nevertheless, applicability of lidars for cloud profiling is limited by the strong attenuation of the lidar signal by optically thick clouds, which often hampers multi-layer and mixed-phase cloud observations.

In 2016, the instrumentation of the French-German Arctic Research Base AWIPEV at Ny-Ålesund, operated by the Alfred Wegener Institute Helmholtz Centre for Polar and Marine Research (AWI) and the French Polar Institute Paul Emile Victor (PEV), was complemented with a cloud radar and now has state-of-the-art instrumentation for vertically resolved cloud

observations. Within the Transregional Collaborative Research Center (TR 172) "Arctic Amplification: Climate Relevant Atmospheric and Surface Processes, and Feedback Mechanisms (AC)[3]" (Wendisch et al., 2017) comprehensive observations of the atmospheric column have been performed at the AWIPEV station at Ny-Ålesund since June 2016.

In this study the vertical structure of clouds at Ny-Ålesund is characterized for the first time using the lidar-radar synergy. In particular, we used the cloud radar to get information about the cloud structure through the whole atmospheric column. Instru-

mentation, data products and models used in this study are presented in section 2. Thermodynamic conditions at Ny-Ålesund for the investigated time period are shown in section 3. In section 4 the vertical hydrometeor distribution and occurrence of different cloud types at Ny-Ålesund are analyzed. For single-layer clouds, which can be liquid, ice and mixed-phase, liquid and ice water path (LWP and IWP) are derived and discussed. The cloud occurrence is related to the thermodynamic conditions such as temperature and humidity (section 4). Since environment temperature and humidity are one of the main parameters

affecting cloud formation and development characterization in models, in section 5 we compare the observed relations between the cloud occurence and thermodynamic conditions with those produced by the numerical weather prediction (NWP) model ICON (Icosahedral Non-hydrostatic; Zängl et al., 2015). Finally, the discussion of results and the summary are given in section 6 and the outlook in section 7.



## 2 Instrumentation and data products

In this study we use various measurements and products continuously running at the AWIPEV observatory. A set of passive and active remote sensors provides information about the thermodynamic state and, cloud and precipitation profiling. In the following subsections, we give an overview of the instruments, data products and model data as well as a short description of the measurement principles and retrieval methods. Table 1 summarizes the measured quantities and retrieved parameters of the instruments and table 2 gives an overview on the Cloudnet products used for the cloud analysis.

### 2.1 Radiosonde observations

Radiosondes have been launched at AWIPEV at least once per day at 11 UTC for more than two decades (Maturilli and Kayser, 2016a). The radiosondes provide vertical profiles of temperature, humidity, wind speed and wind direction. From 21 of May 2006 to 2 May 2017 all radiosondes were of the type Vaisala RS92 and have been processed using the GRUAN version 2 data processing algorithm (Sommer et al., 2012; Maturilli and Kayser, 2016a). The processing corrects for errors in temperature and humidity, for instance, temperature uncertainties due to the heating effect by solar radiation and for humidity errors due to a radiation dry bias (Dirksen et al., 2014). Dirksen et al. (2014) reported that after the GRUAN processing the uncertainties of temperature are 0.25 K and 0.15 K for day and night time, respectively, and 4% for relative humidity at altitudes up to 10 km.

In the present study we use the radiosonde data to characterize the thermodynamic state at Ny-Ålesund for the period from June 2016 to July 2017. In addition, we compare atmospheric parameters of the analyzed period with a previous 23-year long homogenized radiosonde dataset (Maturilli and Kayser, 2016b, 2017). To relate cloud properties to the thermodynamic conditions under which they occur, we combined the radiosonde data with measurements from ground-based instruments operated by AWIPEV.

### 2.2 Microwave radiometer

At AWIPEV, passive microwave observations have been performed with the humidity and temperature profiler (HATPRO; Rose et al., 2005) since 2011. HATPRO is a fourteen-channel microwave radiometer that measures atmospheric brightness temperatures (TB) at K-band (22.24-31.40 GHz) and at V-band (51.26-58 GHz) frequencies with a temporal resolution of 1-2 s. The six K-band channels (22.24 GHz, 23.04 GHz, 23.84 GHz, 22.44 GHz, 26.24 GHz, 27.84 GHz) are located close to the water vapor absorption line at 22 GHz. The 31.4 GHz channel is located in the atmospheric window. The TBs measured at K-band are used for integrated water vapor (IWV), LWP and humidity profile retrievals. The seven V-band channels (from 51.26 GHz to 58 GHz) are located along the oxygen absorption complex at 60 GHz and are used for vertical temperature profiling.

A multi-varied linear regression algorithm (Löhnert and Crewell, 2003) was applied to the TB observations to derive LWP and IWV as well as temperature and humidity profiles. In order to determine the site specific regression coefficients a dataset of almost 3800 Ny-Ålesund radiosondes was used in combination with a radiative transfer model (RTM) to simulate the HATPRO



TBs. In addition to GRUAN processing all the radiosonde data were quality controlled according to Nörenberg (2008). To this end, radiosondes that did not reach the 30 km height were extended with climatological profiles.

In order to determine and correct for potential TB offsets, we followed the method by Löhnert and Maier (2012). The assessment of the TB offsets allows reducing systematic errors in TBs originated from instrumental effects as well as from

radiative transfer simulations. For this method, only clear sky cases were used. In order to identify clear sky situations, i.e. cases without any liquid, HATPRO zenith measurements were checked within 20 minutes before and after a radiosonde launch. In particular, we checked standard deviations of the retrieved LWP within every 2 minutes. If all the standard deviation values of the LWP within 40 minutes did not exceed 1.2 g m$^{-2}$, these cases were considered as clear sky. The TBs measured by HATPRO were then compared to the TBs simulated from the radiosonde data and mean TB offsets were determined. In this

way, a TB offset correction was performed for each period between two absolute calibrations of the instrument.

For this study, we used the retrieved LWP from HATPRO to get information about the amount of liquid in the atmospheric column. This LWP information is also used in the Cloudnet product, which is presented in subsection 2.2.5. The typical uncertainty for the LWP retrieved from HATPRO measurements is 20-25 g m$^{-2}$ (Rose et al., 2005). HATPRO measures continuously during the whole day but can not provide reliable information during rain conditions when the radome of the

instrument is wet. In these cases, data have been flagged and excluded from the analysis.

## 2.3 Ceilometer

Since 2011 a Vaisala ceilometer CL51 has been operated at the AWIPEV observatory (Maturilli and Ebell, 2018). The ceilometer emits pulses at 905 nm wavelength and measures atmospheric backscatter with a temporal resolution of about 10 s and a vertical resolution of 10 m. The maximum profiling range is 15 km.

The ceilometer is sensitive to the surface area of the scatterers and is thus strongly affected by high concentrations of particles like cloud droplets and aerosols (O'Connor et al., 2005). On the one hand, it is thus well suited to detect liquid layers and cloud base heights. On the other hand, the near-infrared signal is significantly attenuated by liquid layers. Therefore, the ceilometer often cannot detect cloud particles above the lowest liquid layer when optical depth exceed a value of around 3.

Protat et al. (2006) reported that ceilometers are essential for reliable detection of high level ice clouds. However, a lidar

system alone is not sensitive enough to detect clouds with low ice water content in the order of less than $10^{-6}$ g/m$^3$. (Bühl et al., 2013). In this study, the attenuated backscatter profiles of the ceilometer CL51 are used in the Cloudnet product (see subsection 2.2.5).

## 2.4 94 GHz radar

On 10th of June 2016 a new 94 GHz cloud radar (JOYRAD-94) of the University of Cologne was installed at AWIPEW

station. JOYRAD-94 is a frequency-modulated continuous wave (FMCW) Doppler W-band radar. The active part of the radar measures at 94 GHz. The radar also has a passive channel at 89 GHz which is well suited for LWP retrievals. Küchler et al. (2017) showed the details on the operational principle and signal processing for JOYRAD-94. At Ny-Ålesund JOYRAD-94 is operated in a high-vertical resolution mode (Küchler et al., 2017). The temporal resolution of the cloud radar is 2.5 s, the





vertical resolution is changing with a height from 4 to 17 m. Minimum detection height was 100 m above ground. Table 1 shows the main settings and parameters for the high-vertical resolution mode. JOYRAD-94 data is available from 10th of June 2016 to 26th of July 2017 when it was replaced by a similar instrument MIRAC-A. In this study, we restrict the analysis to the first year of measurements when JOYRAD-94 was operating. Profiles of the radar reflectivity factor and the mean Doppler

velocity were used in the Cloudnet algorithm to provide information on cloud boundaries, cloud phase and microphysics.

## 2.5   Cloudnet products

The Cloudnet algorithm suite (Illingworth et al., 2007) combines observations from a synergy of ground-based instruments. Cloudnet output includes several products such as a cloud target classification and products with microphysical properties (e.g. ice water content (IWC), liquid water content). In order to provide the full vertical information on clouds, Cloudnet requires

measurements of a Doppler cloud radar, a ceilometer/lidar, a microwave radiometer and thermodynamic profiles of a NWP model. For Ny-Ålesund, measurements are taken from the 94 GHz FMCW cloud radar JOYRAD-94, the ceilometer CL51, and the HATPRO MWR. Model data are taken from GDAS1 (Global Data Assimilation System) or NWP ICON that will be presented in the next subsection. For the first time, data from a FMCW cloud radar with a varying vertical resolution was implemented in the Cloudnet algorithm. Within Cloudnet, the measurements are scaled to a common temporal and vertical

grid of 30 s and 20 m, respectively.

For the target classification the lidar backscatter and Doppler radar parameters are analyzed in combination with thermodynamic profiles of a model (Hogan and O'Connor, 2004). As an example, measurements from the radar and the ceilometer, and the Cloudnet target classification on 29 September 2016 are shown in Fig.1. The target classification consists of the categories such as 1) aerosols and insects, 2) insects, 3) aerosols, 4)melting ice and cloud droplets, 5) melting ice, 6) ice and supercooled

droplets, 7) ice, 8) drizzle/rain and cloud droplets, 9) drizzle or rain, 10) cloud droplets only, and 11) clear sky. In this study, the Cloudnet target categorization is used to differentiate cloud phase (liquid, ice and mixed-phase) and to identify different cloud types.

Based on the target classification, various cloud microphysical retrievals are applied within Cloudnet. The Cloudnet IWC product, which is used in this study, is based on a Z-IWC-T relation (Hogan et al., 2006; Heymsfield et al., 2008), where Z is

a radar reflectivity factor and T is a temperature. The Cloudnet IWC has a bias error and typical random error of 0.923 dB and 1.76 dB, respectively. Heymsfield et al. (2008) found that uncertainties of the IWC retrieval differ for different temperature ranges and are estimated to be +100 %/-30% for temperatures below -40° and about 50 %/-33% for temperatures above -20°. More detailed information on the Cloudnet products can be found in Illingworth et al. (2007).

For the analyzed period (June 2016 - August 2017), the Cloudnet availability is more than 90 % availability for most of the

months (Fig.2). Exceptions are June 2016 (installation of the radar on 10th June), July 2016 and February 2017 (new software installation for cloud radar) with a data coverage of 64 %, 85 % and 81 %, respectively. The total number of analyzed Cloudnet profiles is 1,130,030 that include 216,860 clear sky profiles and 913,170 cloudy profiles.



## 2.6 Model data

### 2.6.1 Global data assimilation system GDAS1

The Global Data Assimilation System (GDAS; Kanamitsu, 1989) is operated by the National Weather Service's National Centers for Environmental Prediction (NCEP). This system analyzes different type of observations and maps the results on a grid used for model initializations. The GDAS data are generated at 00, 06, 12 and 18 UTC. The program produces a 3 hour dataset on 23 pressure levels.

In the present study the GDAS1 data (see https://www.ready.noaa.gov/gdas1.php for detailed information), which are available on a 1° by 1° latitude-longitude grid, was used in the Cloudnet algorithm to provide thermodynamic information for the period from 10 June 2016 to 31 January 2017. The vertical resolution varies from 173 m near the ground to ∼2.5 km at the height of 15 km.

### 2.6.2 NWP ICON model

The ICOsahedral Non-hydrostatic (Zängl et al., 2015) modeling framework for global NWP and climate modeling is developed by the German Weather Service and the Max Planck Institute for Meteorology. The grid structure of ICON is based on an icosahedral (triangular) grid with an average resolution of 13 km. The averaged area of the triangular grid cells is 173 km$^2$. In the vertical dimension, the model has 90 atmospheric levels up to the maximum height of 75 km. The vertical resolution ranges from 30 m at the lowest heights to about 500 m at about 15 km height. For this study we used a column output for Ny-Ålesund taken from the operational global ICON model run. In particular, vertical profiles of environment temperature and humidity, specific cloud water content, specific cloud ice content, rain mixing ratio and snow mixing ratio were used in this study. The ICON column output for Ny-Ålesund is available since 1 February 2017 and has been used as an input for the Cloudnet algorithm since then.

In this study, we use ICON data to exemplarily show how such an observational dataset of clouds can be used for a model evaluation (section 5). We relate the occurrence of different types of clouds in the ICON to temperature and humidity and compared the results to the observational statistics. ICON model output for Ny-Ålesund is available twice a day at 00 UTC and 12 UTC with a forecast for 7.5 days (180 hours) and hourly output intervals. The data only from the first 12 hours after the initialization of the model run were used in our analysis.

## 3 Thermodynamic conditions

It is well known that environmental temperature and humidity strongly influence cloud formation and development. Therefore, we start our analysis with an insight into the thermodynamic conditions during the study period. In this section we also check how representative this time period is in terms of thermodynamic conditions in comparison to the long-term mean.

Figure 3a shows the monthly mean atmospheric temperature based on the radiosonde data for the period from June 2016 to July 2017. The atmospheric temperature follows an annual cycle typical for the Northern hemisphere with higher temperatures





during summer and autumn and lower temperatures in winter and spring. The lowest values of the monthly mean temperature in the lowest 50 m were observed in March and January 2017 (-11° and -10°C, respectively). The highest observed monthly mean temperature was +7°C in July 2016. Looking at the monthly mean temperature at 5 km height, the minimum and the maximum values of -41° and -17°C were found in January and July, respectively. Despite the fact that Ny-Ålesund is located at

the coastline where the climate is supposed to be milder due to the impact of the ocean, the monthly mean temperature changes by 19° and 24°C in the lowest 50 m and 5 km height, respectively. This large amplitude of the temperature change at Ny-Ålesund can be explained by the regular occurrence of polar day and polar night. When a polar night begins in the beginning of October, atmospheric temperature dramatically decreases and it starts to increase again in late March (Fig.3a). Moreover, the smaller temperature variance at lower altitudes might be related to processes between the surface and the atmosphere and

the conserved energy near the ground.

Figure 3b provides information on monthly mean absolute humidity from June 2016 to July 2017. In summer the water vapor is mostly concentrated in the lowest 1.5 km with the highest monthly mean values of up to 6 g m$^{-3}$ in July 2016 and July 2017. The water vapor in this altitude range is thus the main contributor to the integrated water vapor (IWV). In winter, the monthly mean absolute humidity is much lower with a minimum value of ∼1.5 g m$^{-3}$ (January) in the lowest 1.5 km.

In terms of relative humidity with respect to water (RH$_w$, Fig.3c), it can be seen that the monthly mean RH$_w$ is highest in the lowest 2 km of the atmosphere. This is in agreement with Maturilli and Kayser (2016a). There is no strong seasonal variability of the monthly mean RH$_w$ at altitudes higher than 4 km. In the lower troposphere, the monthly mean RH$_w$, following the temperature cycle, is higher in summer and autumn (ranging from 60 to 94 % in the lowest 2 km) and lower in winter and spring (ranging from 52 to 81 % in the lowest 2 km), except for March 2017. In March 2017, the coldest month in the period of

this study, the monthly mean absolute humidity was relatively low (1.7 g m$^{-3}$) in the lowest 1.5 km while monthly mean RH$_w$ was up to 85 % (Fig.3c). In summer and autumn months high values of monthly mean RH$_w$ occur from the surface to 1.7 km. In winter and spring, the atmospheric layer near the surface is drier and high values of RH$_w$ appear from 0.3 to 1.5 km.

In order to determine if and in which way the thermodynamic properties were special for the study period monthly mean tropospheric temperature anomalies are presented in Fig. 4a. These anomalies have been calculated with respect to the previ-

ous 23 years (1993-2015). Figure 4a shows that, compared to the long-term mean, temperatures are higher for some particular summer months. For example, July 2016 and June 2017 were warmer throughout the whole troposphere with maximum temperature anomalies of up to 2° and 4°C, respectively. Winter months were slightly warmer, too: the difference in atmospheric temperature was up to 2°C in December 2016 and February 2017. January 2017 was much colder with a temperature difference of down to -5°C. In comparison to the previous 23 years, atmospheric temperatures in March 2017 were higher in the

upper troposphere (up to 2°C) and lower (-2°C) in the lowest 1.5 km. The largest positive temperature difference was found for autumn 2016, especially for October 2016 with maximum temperature differences of up to +8°C. Johansson et al. (2017) have already shown that moisture intrusions from the North Atlantic can cause significant local warming in some regions of the Arctic that can reach up to 8 K. In addition, Overland et al. (2017) analyzed the variability of the near-surface air temperature (at 925 mb level) in the Arctic for the period from October 2016 to September 2017. The authors reported that there was an

extreme temperature anomaly exceeding 5°C in the autumn 2016 that is in agreement with our results. Moreover, the authors



showed that this extremely high temperature anomaly was associated with a persistent and unusual pattern in the geopotential height field that separated the polar vortex in the central Arctic into two parts. This situation lead to southerly winds which transported warm air into the Arctic from the mid-latitude Pacific and Atlantic oceans (Overland et al., 2017).

The observed anomalies in the monthly mean absolute humidity and IWV (Fig. 4b) in principle follow the sign of the discussed temperature anomalies. Fig.4b shows a correlation between the temperature and IWV increase. For instance, months that have a positive temperature difference also have an increase in the absolute humidity and IWV. Negative temperature differences correspond to decreases in the absolute humidity. For example, January 2017 was particularly colder and drier with anomalies in absolute humidity and IWV of $\sim$-0.5 g m$^{-3}$ and $\sim$-0.8 kg m$^{-2}$, respectively.

Higher IWV values in comparison with the previous years were observed in June 2016, autumn 2016, December 2016 and July 2017. The differences in IWV varied from 1 to 5 kg m$^{-2}$ with largest contributions from the lowest 3 km. In October 2016, the absolute humidity anomaly was highest ($\sim$2 g m$^{-3}$) in the lowest 3 km. This led to a positive change in IWV of more than 5 kg m$^{-2}$ in comparison with previous years.

Thus, it turns out that the period of our study had specific features especially for some months. Maturilli and Kayser (2016a) have shown that in general a significant warming of the atmospheric column at Ny-Ålesund is observed in January and February. The authors reported that this warming in winter is related to the higher frequency of large-scale flow from south-southeast and less from the north. However, in our study January 2017 was much colder in comparison to the previous years. In January 2017, and also in the other winter months, the wind direction occurred more frequently from south-southwest (not shown) in comparison with the earlier period from 1993 to 2014 with wind direction dominated from southeast. However, it is not clear yet what exactly caused the relatively cold January 2017.

## 4 Results

### 4.1 Hydrometeor occurrence

From June 2016 to July 2017, cloudy profiles occur around 80 % of the time (Fig.5). The frequency of cloud occurrence is largest in October 2016 and June 2016 ($\sim$92 %) and lowest in April 2017 (68 %). In order to have a closer look on which types of hydrometeors occur in the atmospheric column, Fig. 5 also gives an overview of the frequency of occurrence of liquid and ice hydrometeors, separately.

For this statistics we check all the range bins in Cloudnet profiles for hydrometeor types. If a Cloudnet bin contains cloud droplets, rain or drizzle we count it as liquid. If ice particles have been detected in a range bin, then we define it as ice. Note that Cloudnet does not distinguish between snow and cloud ice. Mixed-phase range bins are considered as both liquid and ice. Then profiles that contain at least one "liquid" ("ice") bin are counted as liquid (ice) containing. Profiles containing liquid and ice phases are counted in both classes.

Liquid hydrometeors (dashed black line in Fig.5) have the highest frequency of occurrence during summer and autumn (70-80 %) and the lowest in winter ($\sim$36 %). A pronounced seasonal variability is thus visible. Ice (dotted black line in Fig.5) occurs more often in autumn, winter and early spring with the frequency of occurrence varying from 72 to 88 %. In summer




ice occurs typically around 58-78 % of the time. The frequency of ice occurrence does not show a clear seasonal variability as the liquid phase.

Figure 6 shows vertical distributions of hydrometeors. For these statistics we used the above-mentioned bin classification. The frequency of occurrence at a certain altitude was normalized to the total number of Cloudnet profiles in a corresponding

month. The highest frequency of occurrence was 60 and 70 % in March 2017 and October 2016, respectively (Fig.6a, left panel). The lowest frequency of occurrence was in July 2016 ($<$30 %) while for the other months in summer 2016 the frequency of occurrence of all hydrometeors was around 60 %. In January 2017 the occurrence of clouds above 3 km was less than 10 % which correlates with low $RH_w$ (Fig. 3c) at these altitudes and the lowest value of IWV (Fig. 4b).

The total vertical distribution (Fig. 6, right panel, solid black line) shows that hydrometeors occur predominantly in the lowest

2 km with the maximum frequency of occurrence of $\sim$53 % at the height of 660 m. Above 2 km, the frequency of occurrence is less than 30 % and monotonically decreases with height. In terms of seasons, the vertical frequency of occurrence of all hydrometeors reveals variations of the maximum within $\pm$10 % with highest values of frequency of occurrence in autumn 2016 of more than 60 % ($\sim$1 km height). In summer 2016, the hydrometeor frequency of occurrence is in general higher than in summer 2017 indicating a pronounced year-to-year variability which will be analyzed in future when multi-year datasets

will be available.

Liquid hydrometeors (Fig. 6b) occur most of the time in the lowest 2 km. Above 2 km, the frequency of occurrence of liquid is less than 5 % and above 3 km almost no liquid particles are observed. The frequency of occurrence of liquid has a maximum at around 0.7-0.9 km height. Largest values of liquid phase occurrence vary from 40 % to 50 % in summer and autumn 2016. The maximum frequency of occurrence in the winter months does not exceed 15 %. A strong seasonal variability of liquid,

with high values in summer (32 %) and lowest values in winter (12 %) can be seen.

The vertical occurrence of ice hydrometeors is shown in Fig.6c. Ice is mostly present at altitudes below 2 km. On average the frequency of occurrence peaks at around 700 m with values of 40 %. In contrast to the ice occurrence anywhere in a column (Fig. 5), which is not showing a strong seasonal variability, the vertical distribution of ice phase shows a pronounced seasonal cycle, in particular in the lowest 2 km. For higher altitudes, the seasonal variability is less pronounced. Above 2 km,

the frequency of occurrence of ice decreases from $\sim$30 % to less than 10 % at 8 km.

Similar to liquid hydrometeors, the frequency of occurrence of ice is highest in the lowest 2 km with values of 60 and 70 % in October 2016 and March 2017, respectively (Fig.6c, left panel). The lowest ice frequency of occurrence is found for the summer months. In July 2016, which is the warmest month during the observation period, the freezing level often reached altitudes up to 2 km and therefore almost no ice was observed below this height. In January 2017 ice rarely occurred at heights

larger than 4 km, which was probably caused by presence of dry air. In the right panel of Fig.6c it can be seen that the highest frequency of occurrence of ice phase is in the lowest 2 km and around 52 % in autumn, winter and spring.

## 4.2   Statistics on different types of clouds

In addition to the occurrence of hydrometeor types, a classification of clouds into single-layer and multi-layer was also made. Single-layer clouds were furthermore separated into liquid, ice and mixed-phase.



For the classification every Cloudnet profile was checked from the top to the bottom for cloud layers. A cloud is defined here as a layer of at least three consecutive cloudy height bins. Based on a number of identified cloud layers we classified single-layer and multi-layer clouds. We considered cases as multi-layer if two or more cloud layers were separated by one or more clear-sky height bins. Figure 7 gives an overview of the cloud type occurrence at Ny-Ålesund for the whole period of

this study. The total occurrence for the whole period (right-most bar) shows 44.8 % (506,253 profiles) of multi-layer and 36 % (406,810 profiles) of single-layer clouds. Among single-layer clouds the most frequent type was mixed-phase, followed by ice and liquid single-layer clouds with the cloud occurrence of 20.6, 9.0 and 6.4 %, respectively. Note that clouds were considered as mixed-phase if ice and liquid phases were both present in the same cloud boundaries regardless whether liquid and ice were in the same range bin or not. This implies that mixed-phase clouds include not only cases with liquid cloud top and ice below

but also cases when both phases (ice and liquid) are present anywhere within the detected cloud layer.

Figure 7 also shows the monthly occurrence of different cloud types. The monthly cloud occurrence, i.e. the sum of all different cloud types, corresponds to the frequency of occurrence of all hydrometeors shown by a solid black line in Fig. 5. As seen for liquid and ice hydrometeors (Fig. 5), the occurrence of single-layer liquid and ice clouds also has a seasonal and monthly variability. About 15 % of single-layer liquid clouds were detected in summer but less than 2 % in other seasons. The

occurrence of single-layer ice clouds was 15-20 % in winter and spring and less than 5 % in other months. Single-layer mixed-phase clouds and multilayer-clouds were present most of the time with typical values of frequency of occurrence of around 20 % and 45 %, respectively. Thus, during most of the time, cloud systems had a complicated structure and/or consisted of both phases, liquid and ice, indicating that they are related to complex microphysical processes. In turn, the observational capabilities of these types of clouds are limited. In particular, in cases of liquid and mixed-phase multi-layer clouds, the partitioning of the

HATPRO LWP into different cloud layers is related to larger uncertainties. A multi-layer cloud classification requires a reliable profiling of liquid layers, which is limited by significant attenuation of lidar signals in the first liquid layer. Radar signals have better propagation through the whole vertical cloud structure in comparison with a lidar. However, the radar reflectivity is often dominated by the scattering from relatively large particles which mask the presence of small particles, like liquid droplets, being present in the same volume. In the case of multi-layer mixed-phase clouds, liquid phase can thus not be reliably detected

based on radar reflectivity alone.

### 4.3    Focus on single-layer clouds and their relation to thermodynamic conditions

Taking into account the above-mentioned limitations of multi-layer clouds observations our further analysis is concentrated only on single-layer cases. For the following analysis of single-layer clouds we also used LWP from HATPRO and the Cloudnet IWC product. We excluded cases when this information was not available. In particular, profiles with the presence of liquid

precipitation and flagged data due to wet HATPRO radome were excluded. The resulting dataset (Fig. 8, lines with circles, stars and diamonds) was thus reduced to 149,960 profiles (37 % of all single-layer profiles) with 65,299 profiles for single-layer mixed-phase clouds, 59,364 profiles for single-layer ice clouds and 25,297 profiles for single-layer liquid clouds only. Nevertheless, with this subset of single-layer clouds we can still capture the monthly variability and thus assume that it is still representative for all single-layer cloud cases.



Comparison of Fig. 8 and Fig. 7 shows that the occurrence variability of liquid and ice single-layer clouds is similar. Occurrence of mixed-phase clouds differs because of the exclusion of liquid precipitation clouds which often contain ice phase and melting layer and thus considered as mixed-phase. The maximum and minimum occurrence of single-layer mixed-phase clouds of 25 % and 4 % were observed in May 2017 and June 2016 respectively. The annual-averaged top-height of single-layer

mixed-phase clouds was 2 km (not shown). Our findings are in a good agreement with space-borne radar-lidar observations of clouds in the Svalbard region in the period from 2007 to 2010 (Mioche et al., 2015). The authors showed that single-layer mixed-phase clouds in the Svalbard region mostly occur in May.

The geometrical thickness of the single-layer clouds is shown in Fig.9a. The thickness of single-layer liquid clouds varies between 60 to 2200 m with mean and median values of 280 and 240 m, respectively. Less than 1 % of observed single-layer

liquid clouds have a thickness larger than 800 m. In contrast, single-layer mixed-phase clouds typically have a larger geometrical cloud thickness which varies from 100 to 8500 m with the median and mean values of 1100 and 1500 m, respectively. In comparison with mixed-phase single-layer clouds, the geometrical cloud thickness distribution for single-layer ice clouds is broader ranging from 60 to 9500 m. The median and mean values of the geometrical cloud thickness for single-layer ice clouds are 1500 and 2100 m, respectively. The mode of the thickness distribution of single-layer ice clouds correspond to 800 m. Less

than 1 % of single-layer mixed-phase and ice clouds have a geometrical cloud thickness larger than 3 and 4.2 km, respectively.

The frequency of LWP occurrence for liquid and mixed-phase clouds is shown in Fig. 9b. Both types of clouds are characterized by relatively low values of LWP. The median values of LWP for single-layer liquid and mixed-phase clouds are 17 and 37 g m$^{-2}$, and mean values are 30 and 66 g m$^{-2}$, respectively. More than 90 % of single-layer liquid and mixed-phase clouds have LWP values lower than 100 and 200 g m$^{-2}$, respectively. It has to be noted that in particular in these LWP ranges, the

relative uncertainty in the retrieved LWP is quite large (see Sec. 22.2). Larger LWP values in mixed-phase clouds might be related to their larger geometrical thickness (Fig. 9a).

Median values of IWP for single-layer ice and mixed-phase clouds are 14.6 and 21.4 g m$^{-2}$, and mean values are 273 and 164 g m$^{-2}$, respectively (Fig. 9c). IWP values exceeding 400 g m$^{-2}$ are more frequent in single-layer ice clouds than in single-layer mixed-phase clouds. However, for both cloud types the occurrence of IWP values higher than 125 g m$^{-2}$ is less

than 3 %.

A number of studies comparing observed and modeled LWP and IWP values for Arctic regions have revealed the challenge for NWP models to accurately simulate LWP and IWP. Tjernström et al. (2008) evaluated 6 regional models and found that one half of the models showed nearly 0 bias in LWP while another half underestimated LWP by ∼20 g m$^{-2}$. The authors reported that some of the models showed -30 to 30 g m$^{-2}$ biases in IWP. In addition a low correlation between the observations and

modeled IWP and LWP was found. Most of the models showed too low variability of IWP. Karlsson and Svensson (2011) compared 9 global climate models in the Arctic region. The authors showed that mean and standard deviations of modeled IWP and LWP can vary by a factor of 2. Forbes and Ahlgrimm (2014) concluded that such discrepancies may be related to an insufficient representation of microphysical processes. Authors note that one of the major challenges are phase partitioning and a parameterization of cloud particle's formation and development.





Klein et al. (2009) compared 26 NWP models with air-borne and ground-based observations. The authors found that although many models showed an LWP exceeding IWP (as observed), simulated LWP values were significantly underestimated. Since climate and NWP models typically parameterize cloud phase as a function of temperature, we thus analyzed relations between temperature and the phase partitioning for mixed-phase clouds at Ny-Ålesund. Figure 10 shows the probability of liquid

fraction, i.e. (LWP/(LWP+IWP)), in mixed-phase clouds for different cloud top temperature ranges based on the Ny-Ålesund radiosonde observations. In general, the liquid fraction increases with cloud top temperature. Thus, high liquid fraction values in single-layer mixed-phase clouds are found at cloud top temperatures ranging from -15° to 0°C. The occurrence of the liquid fraction of 0.4-0.6, implying that both phases are roughly equally present, is relatively high for cloud temperature ranges between -25° and -15°C but is rare for cloud top temperatures below -25°C. Almost no liquid was observed at the cloud top

temperatures below -40°C. Non-zero liquid fraction below -40°C is mostly associated with thick clouds having a high cloud tops with a liquid layers detected at lower altitudes.

In-cloud atmospheric temperature and humidity are important for NWP models as these parameters determine the cloud particle's formation and development. Therefore, in this study we also relate different cloud types to environmental conditions under which they occur. The frequency of occurrence of the different hydrometeors in single-layer clouds as a function of

in-cloud temperature and relative humidity observed at Ny-Ålesund is shown in Fig.11 (a-d). Here, temperature and relative humidity were determined for each cloud bin between cloud boundaries. For this analysis we only used single-layer-cloud profiles observed one hour before and after a radiosonde launch. We assumed that the atmospheric conditions did not change too much within this time period. For temperatures lower than 0°C the relative humidity with respect to ice (RHi) was used. Values of RHw were used at temperatures exceeding 0°C. For the cloud classification we used the method specified in Sec. 4.4.2.

All single-layer clouds were observed in the temperature range from -60° to +10°C (Fig. 11a). In some cases single-layer clouds appeared at low RHi and RHw (Fig. 11 a-d) that might be associated with hydrometeors falling from saturated to subsaturated atmospheric layers. Another reason could be that the radiosondes, which drift, do not provide representative information for the sampling volume of the zenith-pointing ground-based instruments. However, cases with very low relative humidity values occured in less than 1 % of the analyzed observations.

Figure 11b shows that ice clouds mostly occur in the temperature range from -45° to -5°C including the temperature range (<-38 °C) of homogeneous nucleation. The highest occurrence of ice was observed in the temperature range from -25° to -20°C and under conditions that are subsaturated with respect to water but saturated with respect to ice (Fig.11b). Observed ice particles mostly occur at RHis between 100 and 125 %. Presence of ice at positive temperatures might be related to cases of cloud type misclassification, for example, when a cloud was identified as ice instead of mixed-phase. These cases might be

also associated to uncertainties in the model temperature profile used in the classification algorithm.

Mixed-phase clouds were observed at supersaturation with respect to ice (Fig.11c). Most of the cases were located at the water saturation line. Frequently, mixed phase occurs at temperatures from -25° to +5°C with two maxima in the range of -15° to 0°C. The temperatures of -15° and -5° correspond to the highest efficiency of deposition growth of ice crystals at water saturation levels (Fukuta and Takahashi, 1999).



Liquid phase mostly occurs near water saturation at temperatures from -15° to +5 °C (Figure11d). Supercooled liquid was observed at temperatures down to -40°C. The lowest temperature limit for liquid clouds only was -30°C (not shown).

# 5 Application for model evaluation

This observational cloud data set can provide useful information for a model evaluation. As an example, this section presents a comparison of the NWP model ICON with the observations at Ny-Ålesund. Note that the intention here is not to perform a thorough model evaluation but to show the potential of such a dataset to test, for example, if the dependence of the occurrence of clouds on the thermodynamic conditions can be reprocessed by the model.

The statistics on different types of clouds, their phases and the relation to atmospheric conditions provide a useful dataset for a comparison with similar statistics based on the model output.

Based on a $10^{-7}$ kg kg$^{-1}$ threshold in specific cloud water content, specific cloud ice content, rain mixing ratio and snow mixing ratio we identify clouds in the model. We classify the clouds using the same procedure as for the observations (see Sec. 4.2). The value of the threshold in the hydrometeor contents was found empirically: the usage of a lower threshold leads to the higher occurrence of ice clouds in the ICON model which were not identified in observations. For a higher threshold less ice clouds were present in the ICON model than in observations. Right panels in Fig. 11 show the frequency of occurrence of different hydrometeors in single-layer clouds as a function of in-cloud temperature and RHw based on the ICON model data.

Figure 11e shows that modeled single-layer clouds occur within the temperature range similar to the temperatures observed in clouds at Ny-Ålesund (Fig. 11a). Figure 11f indicates that ice clouds in the ICON model typically exist at temperatures from -65° to -5°C. The high occurrence of ice phase in ICON is found at RHi up to 110 %, while for the observations reveal RHi of up to 125 % (Fig. 11b). The presence of ice particles at lower supersaturation over ice in ICON model in comparison with observations may be associated with ice nuclei (IN) information that is specified in ICON model. Since the concentration of ice particles depends on concentration of IN or secondary ice production, the increasing IN concentration results in more ice crystals that deposit IWV to ice. In addition, higher supersaturation in observations could also explain that these ice-related processes are longer in the real cloud than it is specified in ICON model. Thus, the variability of the IN efficiency is still challenging for models (Fu and Xue, 2017). Both, the ICON model and observations, reveal that ice particles typically occur at relative humidities higher than the saturation over ice but lower than the saturation over water.

Mixed-phase clouds in the ICON model appear near the water saturation (Fig. 11g) that is consistent with the observations (Fig. 11c). The model mostly produces mixed-phase clouds within the temperatures range from -10° to +5°C, that is narrower in comparison to the observations.

Modelled liquid phase occurs near water saturation at temperatures from -15° to +5°C (Fig.11h), which is in a good agreement with observations. In the ICON model the occurrence of liquid phase at temperatures below -5°C is only 6 %, while in the observations this occurrence is more than 30 %.

Figure 12 summarizes temperature dependencies of hydrometeor occurrences in the ICON model and in observations. The temperature distributions of single-layer liquid clouds (solid red lines, Fig. 12) are narrow (-10° to +5°C) for both, model





and observations, although, the observed distribution has larger occurrence values. The total distributions of the liquid phase (dashed red lines, Fig. 12) are different. The observed distribution has larger values and occupies a wider temperature range (-25° to +10°C). In the model, most of the liquid phase is concentrated in the temperature range from -10° to +5°C. This difference leads to a divergence between mixed-phase cloud occurrences (solid green lines, Fig. 12): the observed frequency

distribution for mixed-phase clouds shows a broader temperature range than the model. Sandvik et al. (2007) showed a similar difference between observed and modelled single-layer mixed-phase clouds. For the modelling the polar version of the nonhydrostatic mesoscale model from the National Center for Atmospheric Research was used. The authors found that for temperatures below -18° the liquid fraction in single-layer mixed-clouds was completely absent in simulations.

Ice cloud observations (solid blue line in a) show a broad temperature range from -60° to +5°C. In comparison to the

observations, the model (solid blue line in b) shows a broader temperature range for single-layer ice clouds (-70° to +5°C). Due to the low occurrence of the liquid phase at temperatures below -5°C by the model most of the clouds at lower temperatures are classified as pure ice. Therefore, the model shows significantly larger occurrence of ice clouds at temperatures warmer than -20°. Also this explains similarities between modelled ice phase in pure ice and ice-containing clouds (dashed blue line). In addition, the occurrence of simulated ice clouds is higher at temperatures below -40°, which corresponds to the homogeneous

ice nucleation regime.

## 6   Summary and discussion of results

This study provided, for the first time, a statistical analysis of clouds at Ny-Ålesund, Svalbard and their relation to the thermodynamic conditions under which they occur. We analyzed a 14-month long measurement period at Ny-Ålesund and presented statistics on vertically resolved cloud properties, hydrometeors and thermodynamic conditions. The Cloudnet categorization

scheme, based on observations from a set of ground-based remote-sensing instruments (active and passive), was applied in order to provide vertical profiles of clouds, their macrophysical, microphysical properties and phase. In total 1,130,030 Cloudnet profiles are available for the period from June 2016 to July 2017.

The statistics on cloud properties and atmospheric thermodynamic conditions is essential for a better understanding of cloud processes and can also be used for model evaluation. In this study, the relation between cloud properties and thermodynamic

conditions from observations was compared to results from the NWP ICON model.

The thermodynamic conditions were derived from radiosonde data for the period from June 2016 to July 2017 and were compared with previous 23 years. This comparison revealed that the period of our study differs from the previous years. January 2017 was significantly colder with temperature differences down to -5°C while October 2016 was extremely warm with temperature anomalies of more than +5°C. Also, in comparison to the previous 23 years, IWV was lower in January 2017

by 1 kg m$^{-2}$ and more than 5 kg m$^{-2}$ higher in October 2016.

The main findings and their discussions are listed below.

(1) The total occurrence of clouds is ~81 %. The highest frequency of occurrence is in October 2016 (92 %). Similar results of high cloud occurrence in summer and autumn at Ny-Ålesund based on micro-pulse lidar measurements were previously



found by Shupe et al. (2011). Nevertheless, the observed total occurrence of clouds at Ny-Ålesund for the investigated period is higher than the one from Shupe et al. (2011). The authors showed that the total annual cloud fraction at Ny-Ålesund for the period from March 2002 to May 2009 was 61 %. On the one hand, we analyzed the different time period. On the other hand, the occurrence of clouds in Shupe et al. (2011) might be underestimated when only a lidar is used (Bühl et al., 2013). However,

our results are in a good agreement with a previous study by Mioche et al. (2015). The authors used space-borne observations over the Svalbard region for the period from 2007 to 2010. They applied the DARDAR algorithm (Delanoe and Hogan, 2008, 2010) that utilizes measurements from CALIPSO and CLOUDSAT. They showed that the cloud occurrence over the Svalbard region was in the range from 70 % to 90 % having peaks in spring and autumn. Mioche et al. (2015) found the lowest cloud occurrence in July, while the statistics in the present study reveal the high cloud occurrence (∼80 %) in this month. Also here,

this difference might be related to the different periods investigated. Another reason might be that the observed clouds in July are predominantly located at heights below 1.5 km. These low-level clouds are difficult to capture by CloudSat due to its "blind zone" in the lowest 1.2 km (Marchand et al., 2008; Maahn et al., 2014). Mioche et al. (2015), for example, showed that the Ny-Ålesund ground-based measurements revealed the highest cloud occurrence in summer (between 60 % and 80 %), while satellite observations showed the minimum in that season. The lowest cloud occurrence in the study by Shupe et al. (2011) is

around 50 % in March. In our study, the lowest cloud occurrence (∼65 %) was also observed in spring. This might be associated with a relatively low atmospheric temperature and less moisture being available in the atmosphere. The increase of cloudiness in summer and autumn is probably due to higher values of relative humidity at the site in comparison with other seasons. Also sea-ice coverage might impact the cloud occurrence. As during summer and autumn sea ice coverage is the lowest, areas of open water are larger and therefore, that can lead to enhanced evaporation and latent heat exchange with the Arctic atmosphere.

(2) We found that multilayer and single-layer clouds occur 44.8 % and 36 % of the time, respectively. The most common type of single-layer clouds is mixed-phase with a frequency of occurrence of 20.6 %. The total occurrences of single-layer ice and liquid clouds are 9 % and 6.4 %, respectively. The cloud occurrence of single-layer liquid and ice clouds has a pronounced month-to-month and seasonal variability.

  The analysis of cloud phase shows that liquid is mostly present in the lowest 2 km with the highest occurrence in summer

and autumn (especially, in October 2016) and lowest in winter. However, in winter the occurrence of liquid hydrometeors is still significant and reaches 12 % at a height of 1 km. The occurrence of ice phase within the first 2 km is lowest in summer (22 %) and highest in October 2016 and March 2017 with 60 % and 70 %, respectively. The largest frequency of occurrence of ice and liquid in October 2016 (> 50 %) is related to strong temperature and humidity anomalies in this month. According to Overland et al. (2017), the anomalies were associated with warm air transported into the Arctic from mid-latitudes, the Pacific

and Atlantic oceans.

  (3) We analyzed 149,960 Cloudnet profiles with single-layer clouds only. Single-layer liquid and mixed-phase clouds typically have very low values of LWP with median values of 17 and 37 g m$^{-2}$ and mean values of 30 and 66 g m$^{-2}$, respectively. It has to be noted that these low values of LWP may significantly affect shortwave and longwave radiation (Turner et al., 2007). The LWP of single-layer mixed-phase clouds is larger than for single-layer liquid clouds. This result is in an agreement with




a study by Shupe et al. (2006). The authors reported that the LWP for mixed-phase single-layer clouds is larger than for pure liquid clouds due to thicker liquid layers in mixed-phase clouds.

Turner et al. (2018) showed that in Barrow the occurrence of single-layer mixed-phase clouds is lower than the one of single-layer liquid only cloud at LWP values exceeding 120 g m$^{-2}$. At LWP values below 120 g m$^{-2}$ liquid only clouds become dominant over mixed-phase clouds. We found a similar behavior at Ny-Ålesund but with the transition at 50 g m$^{-2}$.

The IWP statistics shows that in general single-layer ice clouds contain more ice than single-layer mixed phase clouds with corresponding mean values of 273 and 164 g m$^{-2}$, respectively. The median values of IWP for single-layer ice and mixed-phase clouds are 14.6 and 21.4 g m$^{-2}$, respectively. This difference might be related to the cloud geometrical thickness. On average single-layer ice clouds are thicker than mixed-phase clouds. Single-layer mixed-phase clouds have higher occurrence than ice clouds for IWP values ranging from 25 to 400 g m$^{-2}$. For IWP values exceeding 400 g m$^{-2}$ ice clouds were more frequent than mixed-phase.

Since phase partitioning in cloud models depends on atmospheric conditions, we analyzed relations between cloud top temperature and liquid fraction for mixed-phase clouds. It was found that liquid is present at temperatures down to -40°C. The highest occurrence of liquid phase is at cloud top temperatures ranging from -15° to 0°C.

(4) We analyzed the occurrence of different cloud types at Ny-Ålesund as a function of environment conditions. In addition to observations we also used the ICON model output for these analyses. We found that the temperature distribution of single-layer liquid clouds is narrow with temperatures typically ranging from -10° to +5°C. Similar results are also found for the ICON model. However, the distribution of the liquid phase for mixed-phase clouds is one of the major differences between the model and observations. The observed distribution ranges from -25° to +10°C while in the ICON model liquid phase is concentrated in the temperature range from -10° to +5°C. This difference results in a significant divergence between observed and modelled single-layer ice and mixed-phase clouds. The observed single-layer mixed-phase clouds occur in a much wider temperature range (from -25° to +5°C) than in the ICON model (from -15° to +0°C). Such differences have been previously reported by Sandvik et al. (2007). The authors showed that models can completely miss single-layer mixed-phase clouds below -18°. Observed ice clouds occur at temperatures from -60° to +5°C while the model simulate ice clouds down to -70°C. The occurrence of modeled ice clouds is significantly larger than observed at temperatures warmer than -20°. Due to the lower occurrence of liquid phase in the model at temperatures below -5° modeled clouds are often classified as pure ice. Also the model shows the higher occurrence of ice clouds at temperatures below -40° where homogeneous ice nucleation takes place.

## 7 Outlook

In order to have, more robust statistics and also to account for year-to-year variability long-term observations at Ny-Ålesund are needed. Therefore, the measurements of cloud and thermodynamic profiles are still ongoing at Ny-Ålesund within the (AC)³ project. The aim of this study is to present the results from the first year of observations and to show their potential to provide vertically resolved cloud information.



The statistics on LWP and IWP for single-layer clouds, provided in this study, show that most of the time single-layer clouds at Ny-Ålesund have very low LWP which is within the uncertainty range ($<30$ g m$^{-2}$). In future, the retrievals of LWP can be improved by using the infrared and higher frequencies of the MWR (Löhnert and Crewell, 2003). The information from the 89 GHz passive channel of the FMCW radar and 183, 233 and 340 GHz frequencies of LHUMPRO (low humidity profiler) of

the University of Cologne, which are currently measuring at Ny-Ålesund, can be used for reducing the uncertainty of LWP.

The next step will be to derive cloud microphysical properties such as LWC, IWC and effective radius for different types of clouds. This information is essential for cloud-radiation interaction studies. Therefore, the derived profiles of single-layer clouds and their microphysical properties will be used in combination with a radiative transfer model to calculate the cloud radiative forcing at Ny-Ålesund. In addition, to show the representativeness of derived cloud properties at Ny-Ålesund among

other Arctic sites with similar ground-based instrumentation, our results will be compared with other locations in the Arctic. In order to make such a comparison consistent, similar methods have to be used for deriving cloud microphysical properties and the same time period has to be analyzed.

Information on cloud microphysical properties can be used to test the representation of clouds and their dependency on temperature and humidity in models and therefore, for an evaluation of high-resolution models.

*Data availability.* The radiosonde data were taken from doi: 10.1594/PANGAEA.845373 and doi: 10.1594/PANGAEA.875196. The Cloudnet data are available at the Cloudnet website (http://devcloudnet.fmi.fi/).

*Author contributions.* TN applied the statistical algorithm, performed the analysis, prepared and wrote the manuscript. KE, UL, MM contributed with research supervision, discussions of the results and manuscript review. UL helped to apply retrievals for HATPRO. MM provided long-term radiosonde dataset. CR provided instrumentation data for this study. EO applied the Cloudnet algorithm for Ny-Ålesund.

*Competing interests.* The authors declare that they have no conflict of interest.

*Acknowledgements.* We gratefully acknowledge the support by the SFB/TR 172 "Arctic Amplification: Climate Relevant Atmospheric and Surface Processes, and Feedback Mechanisms (AC)[3]" in sub-project E02 funded by the DFG (Deutsche Forschungsgemeinschaft). We acknowledge the staff of the AWIPEV Research Base in Ny-Ålesund for helping us in operating the cloud radar, launching radiosondes and providing the MWR and ceilometer data. We gratefully acknowledge DWD service for providing the data of the global NWP ICON model

for Ny-Ålesund. We also thank Patric Seifert for providing GDAS1 data for Ny-Ålesund and helpful discussions.





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





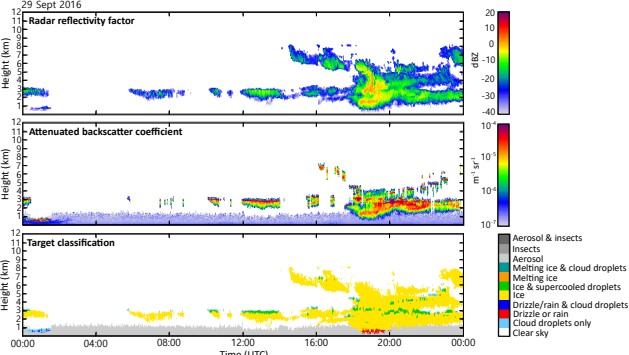

**Figure 1.** Radar reflectivity factor (top), lidar backscatter coefficient (middle) and Cloudnet target classification on 29 September 2016, AWIPEW observatory at Ny-Ålesund.





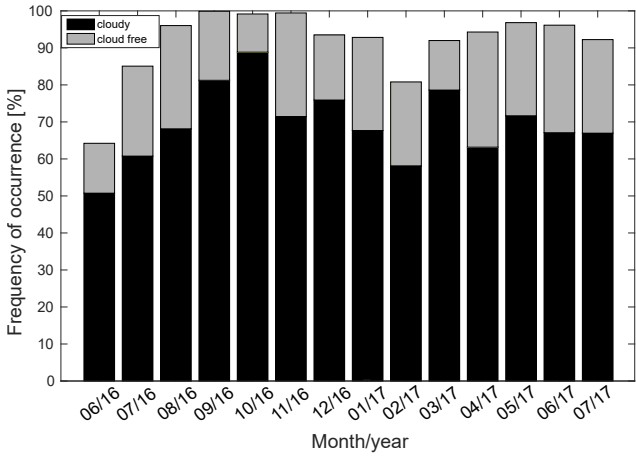

**Figure 2.** Cloudnet data availability for Ny-Ålesund for June 2016 to July 2017. Grey bars correspond to clear sky profiles, black bars to cloudy profiles.





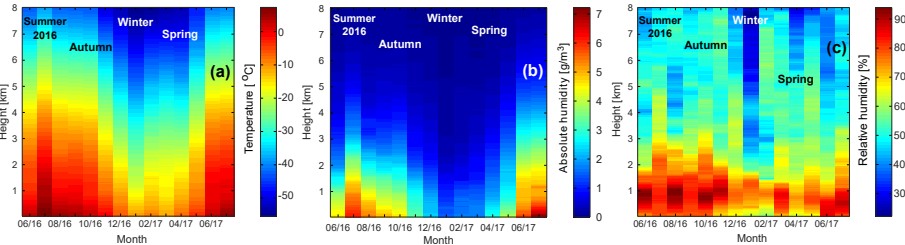

**Figure 3.** Vertical profiles of monthly mean atmospheric temperature (a), absolute (b) and relative humidity (c) from radiosonde observations at Ny-Ålesund from June 2016 to July 2017.





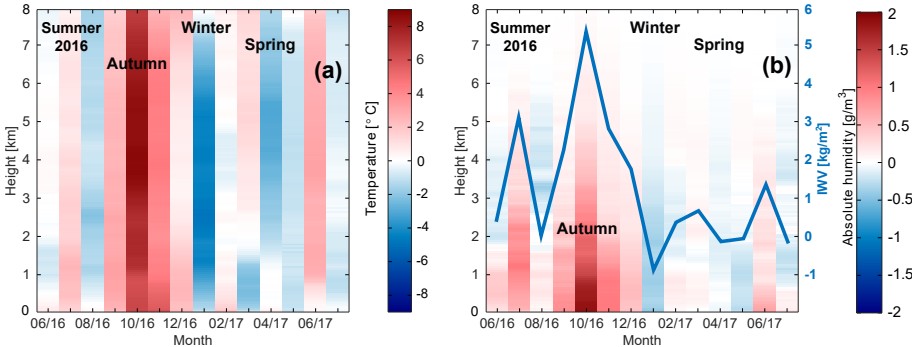

**Figure 4.** Anomalies of monthly mean atmospheric temperature (a) and absolute humidity (b) from radiosonde observations at Ny-Ålesund from June 2016 to July 2017. Anomalies are calculated with respect to the monthly mean values of the previous 23 years (1993-2015). The blue line corresponds to the IWV anomaly for the same time period.



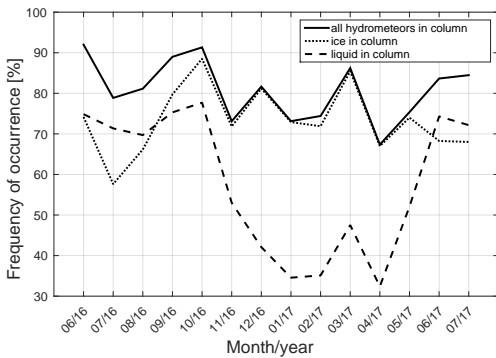

**Figure 5.** Frequency of occurrence of profiles with ice, liquid and any kind of hydrometeors. The frequency is given in % and normalized to the total number of Cloudnet profiles for each month.





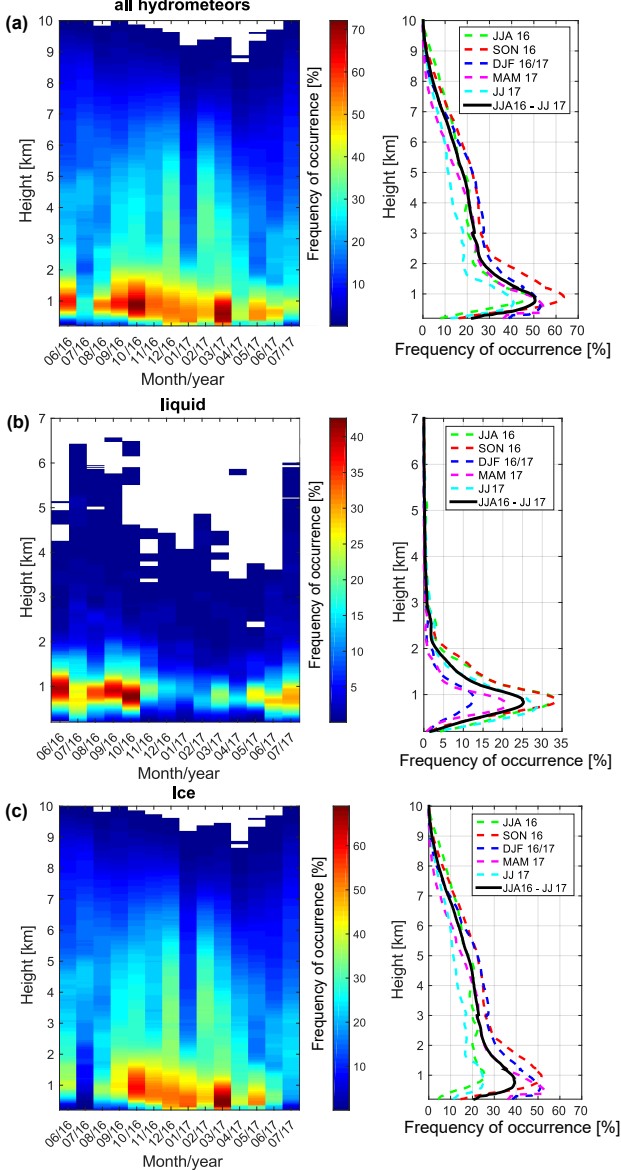

**Figure 6.** Monthly, seasonal and total (for the whole time period) frequency of occurrence of all hydrometeors (a), liquid (b) and ice (c) as a function of height for the period from June 2016 to August 2017. Frequency of occurrence is given in % and normalized to the total number of Cloudnet profiles for each month.





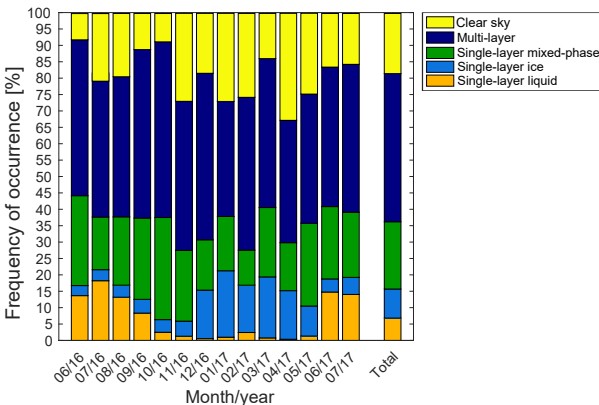

**Figure 7.** Monthly frequency of occurrence of different types of single-layer clouds (liquid, ice and mixed-phase), multi-layer clouds and clear sky profiles for the period from June 2016 to August 2017. Last right column showing the total frequency of occurrence.





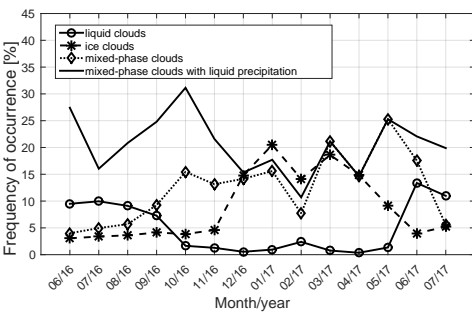

**Figure 8.** Frequency of occurrence of ice-only, liquid-only, mixed-phase single-layer clouds based on Cloudnet categorization data (for lines with circles, diamonds and stars profiles with liquid precipitation are not included). The frequency is given in % and normalized to the total number of Cloudnet profiles in each month.



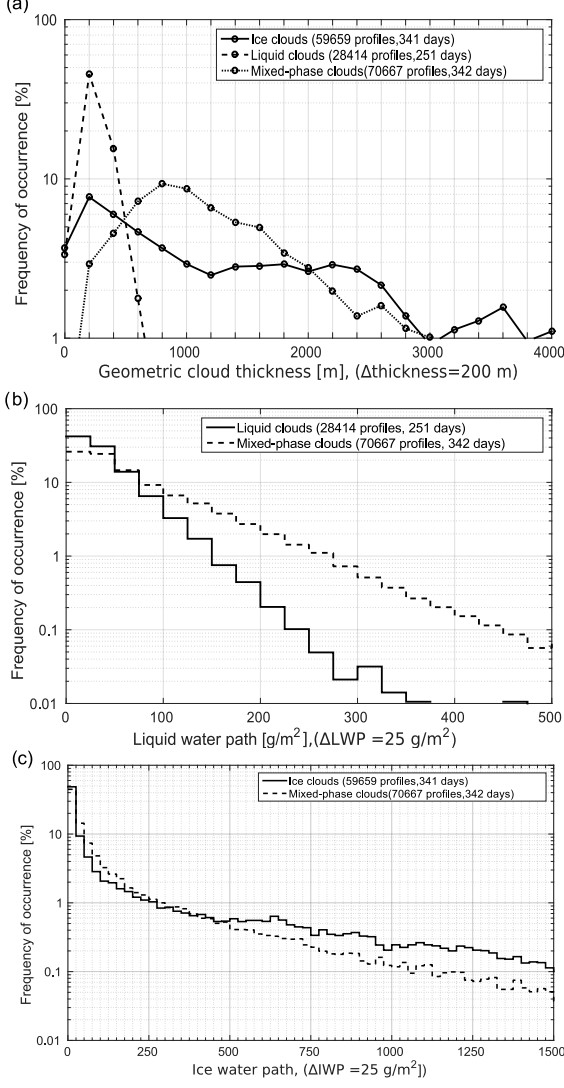

**Figure 9.** Frequency of occurrence of cloud thickness for single-layer clouds (a), of LWP for single-layer liquid and mixed-phase clouds (b) and of IWP for single-layer ice and mixed-phase clouds (c) for the period from June 2016 to August 2017. The y axis is shown in logarithmic scale. Frequency is normalized by the total number of corresponding cloud type.





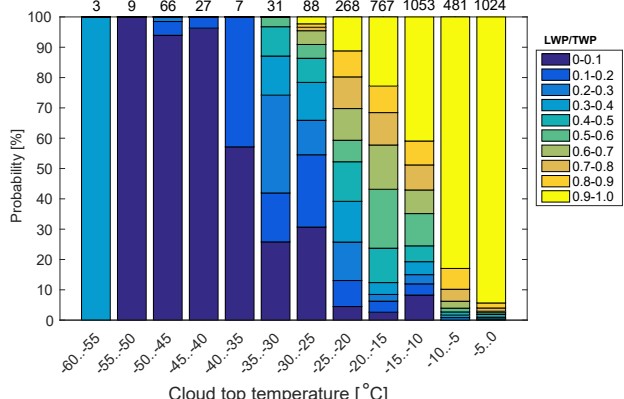

**Figure 10.** The relative probabilities of different ranges of the liquid fraction LWP/TWP given at various cloud top temperatures of single-layer mixed-phase clouds. The probability is normalized by the total number of profiles for each cloud top temperature range. Numbers at the top of plot show the number of cases included in the temperature range. The total number of all profiles is 3824.





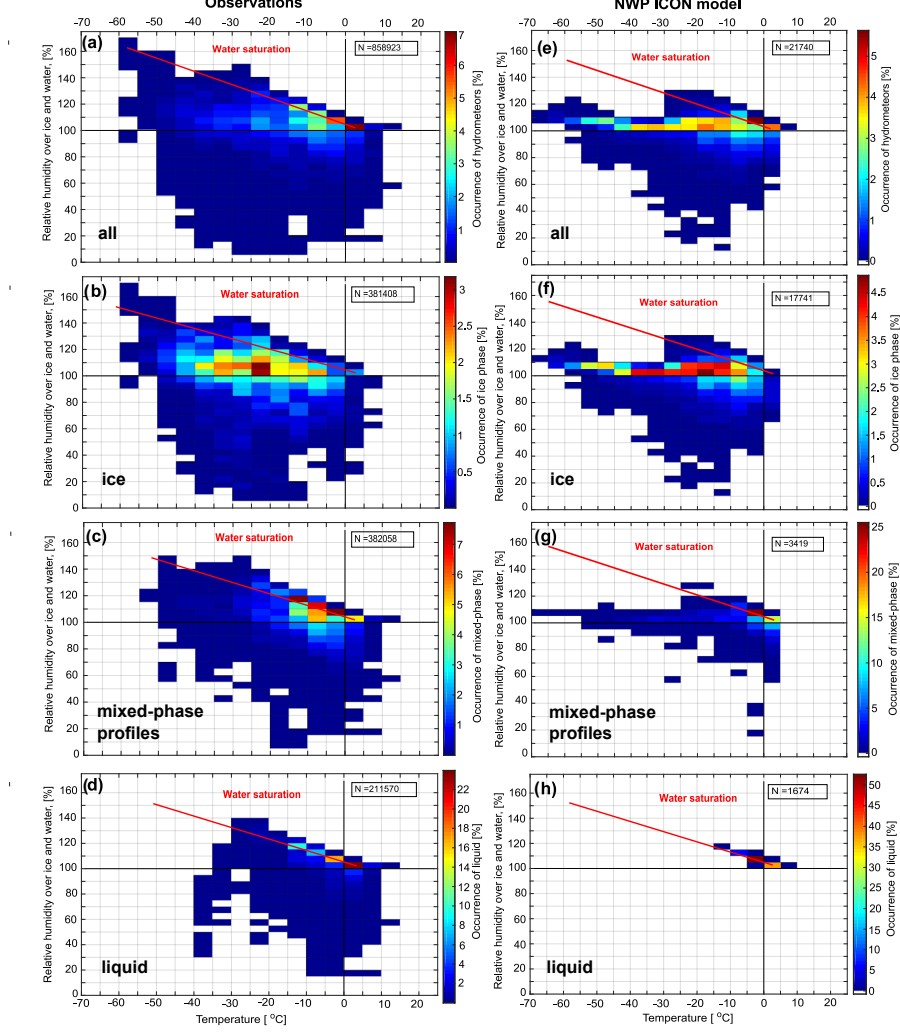

**Figure 11.** Two-dimensional histograms of in-cloud atmospheric temperature and relative humidity for all clouds (a,e), ice clouds (b,f), mixed-phase clouds (c,g). For a, c, e and g ice and/or liquid phases are present. For b and f only ice phase is present. Liquid phase of liquid-containing clouds is shown in d and h. Only cases of single-layer clouds are included and shown for observations (left) and for global NWP model ICON (right). Frequency of occurrence is normalized by the total number of bins of the correspondent single-layer clouds detected between the period of one hour before and after radiosonde launch.





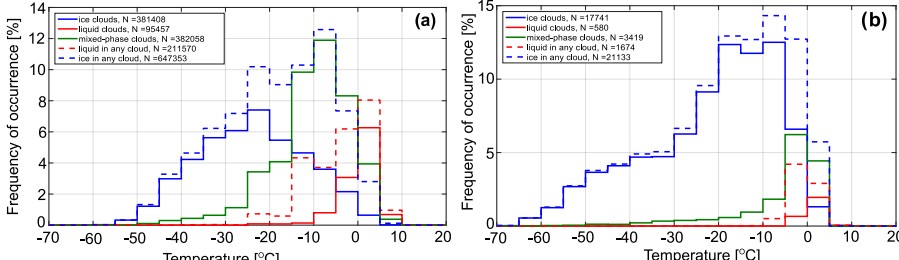

**Figure 12.** Distribution of in-cloud atmospheric temperature for different types of single-layer clouds, liquid and ice phase for observations (a) and global NWP ICON model (b).





**Table 1.** Instruments and data used for this study.

| Instrument | Measured quantities | Settings for Ny-Ålesund | | Retrieved parameters |
| | | Temporal resolution | Vertical resolution (range) | |
| --- | --- | --- | --- | --- |
| MWR HATPRO | Brightness temperatures at 22.24-31.4 GHz (7 frequencies) and at 51.26-58 GHz | 1-2 s | | LWP |
| Ceilometer CL51 | Profiles of attenuated backscatter coefficient | 12-20 s | 30 m | cloud base, liquid layer |
| FMCW 94 GHz cloud Doppler radar (JOYRAD-94) | Profiles of reflectivity (94 GHz) and Doppler velocity (94 GHz), Doppler spectrum width (94 GHz), brightness temperature (89 GHz, passive) | 2.5 s | 4 m (100-400 m) 5.3 m (400-1200 m) 6.7 m (1.2-3 km) 17 m (3-10 km) | cloud presence, cloud boundaries |
| Radiosonde RS92, RS41 | Profiles of atmospheric temperature and relative humidity | at least 1 sonde/day, 1 s | 5-7 m | IWV |



**Table 2.** Cloudnet product used for this study.

| Input parameters (Instrument/model) | Settings for Ny-Ålesund | | Retrieved parameters |
| | Temporal resolution | Vertical resolution (range) | |
| --- | --- | --- | --- |
| LWP (MWR) | 30 s | 20 m | Target classification, IWC product |
| reflectivity factor and Doppler velocity (94 GHz radar) | | | |
| attenuated backscatter coefficient (ceilometer CL51) | | | |
| hourly model analysis and forecasts (GDAS1 or NWP ICON model) | | | |