# Peer review of "Statistics on clouds and their relation to thermodynamic conditions at Ny-Ålesund using ground-based sensor synergy"

_Atmospheric Chemistry and Physics, 2018_

## Referee Comment (RC1) · Anonymous Referee #1 · 13 Nov 2018

This paper presents a 14-month cloud property dataset collected at Ny-Alesund using a multi-instrument retrieval algorithm. The authors the presented some basic statistics to describe the cloud field including: cloud occurrence, column phase information and frequency, dependence on atmospheric state, and more. Generally, this is a well-written paper and I recommend acceptance after some minor items are addressed.

P1,L16: clouds do not conserve LW radiation; "emit" is a better word here P2,L3: mid-summer SW radiative cooling of the surface is not a general feature across the Arctic. You might indicate that clouds cool the surface the entire year at a site like Summit Greenland (Miller et al. 2018) P2,L33: sensors cannot "capture low level clouds" –

"observe" is a better word P4,L29: "multi-variable" P6,L27: I am not sure what these double values (e.g., "+100% / -30%") mean here. Please clarify P7,L3: "US National Weather Service's" P10,L6: I found this minimum in July surprising. I suspect this is just a sampling error due to a relatively small dataset. I would encourage the authors to bring out this point more strongly here and in the conclusions of the paper P11,L20-25: Shupe et al. 2015 also had the same challenges in distributing liquid water vertically in mixed-phase clouds. Does the CloudNet algorithm use the same scaled-adiabatic method? Perhaps mention both the Shupe paper and address this question here. P13,L1: There were not 26 NWP models in the Klein et al. study. Most were very experimental cloud-resolving or LES models. P14,L7: "represented" is a better word than "reprocessed" P14,L10: What sort of radar reflectivity does this represent, given typical hydrometeor sizes? Perhaps a statement can be made here about the advantage of using an instrument simulator? P14,L23: you state "...these ice-related processes are longer in the real cloud than it is specified in ICON model." Does this mean that you believe that the model has the processes act more rapidly than they do in nature? This sentence needs clarification P15,L19: "categorization" → "classification" P16,L33: These clouds with LWP values between 30 and 60 g/m2 also have the largest radiative contribution to the surface energy budget (Bennartz et al. 2013) P18,L3: The L&C paper only talk about higher frequency microwave radiometer measurements. If you want to include the IR here (and you should), then please reference either Marke et al. 2016 or Turner 2007 P18,L6: how would the effective radius be determined? Fig 9, panels B and C: the most important clouds from a radiative point of view are the ones with smaller LWP and IWP. However, the linear x-axis sort of hides them. Would you consider plotting these data on a log10 axis from 1 g/m2 to 1000 g/m2 for LWP? The bins should be equal-sized bins in log10(LWP) space. Ditto for the IWP, but perhaps start the lower range at 0.1 g/m2? Fig11: I don't see the labels for the individual panels (e.g., "a", "b", etc). Also, there seems to be some weird inset artifact on the lower left-hand panel. Fig12: I would emphasize that the ICON results are not "global", but are from output over the Ny Alesund END

---

## Referee Comment (RC2) · Anonymous Referee #2 · 30 Dec 2018

**Review "Statistics on clouds and their relation to thermodynamic conditions at Ny-Ålesund using ground-based sensor synergy" by T. Nomokonova, K. Ebell, U. Löhnert, M. Maturilli, C. Ritter and E. O'Connor**

The manuscript presents a 13 months analysis of vertical cloud information and the corresponding thermodynamic conditions from ground-based instruments and radiosoundings at a research station in Ny-Ålesund, Spitsbergen. The dataset was analysed in terms of cloud occurrence, cloud structure (single layer vs. multi-layer clouds) and cloud phase (liquid, mixed-phase, ice). Additionally the liquid water path and ice water path was estimated. Single-layer clouds were also related to in-cloud temperature and relative humidity. Some of the results were compared to ICON model output. The presented dataset is of great interest and the analysis provides new insights in the structure of Arctic clouds. Some of the analysis could be discussed a bit more thoroughly, therefore I recommend to accept this paper after major revisions.

**1 Major comments:**

- The cloudnet algorithm seams to use thermodynamic profiles of a NWP model. Please explain what that exactly means and how that influences the analysis? If the thermodynamic profiles deviate a lot from real conditions (as measured by radiosounding for example) what are the consequences? And could that probably lead to inconsistency when using in a further analysis later on radiosoundings as additional information?

- Discuss more the uncertainties of the measurements and their role in terms of interpretation of the results, e.g. the uncertainty range of IWC (which seems to be quite large). This could be of importance especially when it comes to comparisons. Include this discussion also in the conclusions when discussing the comparison of observations with ICON model output.

- In section 5 cloudnet products are used in combination with radiosoundings in comparison to ICON model output. I am missing one essential discussion here: How different are the temperature and humidity profiles from ICON compared to the radiosoundings? If the thermodynamic profiles from ICON deviate considerably from the data from radiosoundings the analysis could be limited by that independent of the cloud information derived by cloudnet or from ICON.

- Discuss more what the thermodynamic conditions mean when related to cloud properties. What can be derived from this information and what is still missing? Does not for cloud formation also the stability or layer of the atmosphere (boundary layer, inversion etc.) play a huge role? Could that be analysed with the used dataset as well?

**2 Specific comments:**

- Add the vertical range of each instrument (lowest detection, maximum profilig range). Especially the lowest detection could be interesting to know.

- Page 6, line 27: What is the IWC uncertainty range between -40 and -20°? Please add that information as well.

- Page 7, line 5: Is it not a 6 hour dataset if the data is generated at 00, 06 ... ? Please correct or explain.

- Page 7, line 19: Why was the ICON output not used consistent for the whole time period of analysis? What is the difference introduced by using ICON compared to GDAS1? Which uncertainty results from using one or the other? Explain more what the influence is of the model data used on the cloudnet products and thus on your analysis.

- Page 7, line 24: Why was a forecast product used and not nudged simulations using reanalysis data? Would it not be more correct to use nudged simulations or are there technical limitations? Is the vertical profile data the same as used for Cloudnet?

- Page 9, line 4: How was the monthly mean IWV estimated? From radiosonde profiles?

- Section 3: What is the conclusion from your analysis about the thermodynamic conditions? How much can the results be generalized or what does it mean for the findings of the paper that the year 2017 was different from previous years?

- Page 11, line 29: Please explain how the Cloudnet IWC product is estimated.

- Page 11, line 33/Section 4.3: How does not taking into account cases with liquid precipitation influence the results due to bias for specific cloud types? Or is there no cloud type bias for liquid precipitation?

- Page 12, line 8: How is the geometrical thickness of the single-layer clouds estimated?

- Page 12, line 27: Specify the model domain/region of the 6 regional models.

- Page 13, line 1: Which region/domain was the comparison focused on?

- Page 13, line 22: How would correcting for the drift influence the results or how much uncertainty is introduced by not doing so?

- Page 14, line 20: Explain how the ice nuclei information has to be specified in ICON.

- Page 14, line 21: Why is the IN concentration increasing? Explain your argumentation here better.

- Section 5: Discuss how the model resolution of ICON could influence the results and your interpretation.

- Page 17, line 9: You write that single-layer ice clouds are thicker than mixed-phase clouds. Why is that? Do cloud droplets not increase the cloud thickness?

- Figure 6: It would be interesting to add another subplot of the subfigures b and c showing the profiles for the lower 2 km.

- Figure 8: Why does the liquid precipitation only occur in mixed-phase clouds? Are there no liquid clouds with liquid precipitation?

- Figure 11: How are single-layer clouds defined in ICON? As one model level cloud free in between model level containing clouds? Please specify.

- Figure 11: What is the time resolution of the ICON output used for the analysis compared to the observations? How would the observations look like if not the time period of one hour around the radiosounding would be used but instead the time period of the ICON output? How does the model resolution thus influence the results?

- Figure 11 and 12: You mention here that the global ICON model is used-how different is the resolution compared to the single-column version mentioned earlier? Which domain size/how many grid cells were analysed?

- Figure 12: There is considerably less profiles analysed from the ICON model compared to the observations- how does that influence the analysis? Is the statistics sufficient for general statements about the clouds?

**3  Small remarks,typos:**

- Write out abbreviations in the abstract (AWIPEV, LWP, IWP, ICON).

- The numbering of the subsections is not always correct, e.g. p. 5, l. 12 and p. 5, l. 27 subsection 2.2.5 has to be replaced by subsection 2.5, p. 12 l. 20 Sect. 22.2 does not exist, p. 13 l. 19 Sec. 4.4.2 does not exist.

- The analysed period is not mentioned consistent throughout the manuscript. From the plots etc. it looks like as the analysed period is from middle of June to July 2017. However in the manuscript it is often stated that 14 months are analysed. Also at some place the period is mentioned with different start/end dates, e.g. p. 6 l. 29: August, p. 7 l. 9: 31 January 2017.

- Leave away the article in front of ICON or add model (in ICON ... or in the ICON modell...).

- Some figures (e.g. Fig. 5) seam not to be vectorized and thus have not such a good resolution when zooming in- change if possible.

- Page 1, line 10: delete the (to ICON model output).

- Page 1, line 11-13: Rewrite the sentence/split it into two (Distinct ... evident.).

- Page 1, line 15: Add "budget" (in the energy budget).

- Page 4, line 6: Add model data, which is also included in Table 2.

- Page 4, line 9: Why 21st of May? The analysis starts in June 2006.

- Page 4, line 10: Which radiosonde type was used after 2 May 2017?

- Page 5, line 25: There is a "." too much in front of the citation.

- Page 6, line 27 and page 15, line 13: Add unit after temperature (C).

- Page 7, line 23: Replace compared with compare.

- Page 8, line 5: Do you mean less variable instead of milder?

- Page 8, line 33: Replace K with °C to be consistent.

- Page 11, line 26: Change header to "Single-layer clouds and their relation to thermodynamic consitions".

- Page 11, line 31f: Add percentages for all numbers.

- Page 14, line 1: Replace Figure11d with Fig. 11d.

- Page 16, line 3: Replace "the different time period" with "a different time period".

- Page 17, line 12: Replace "cloud models" with NWP or climate models (depending on what you want to say here).

- Page 17, line 20 and 22: One time you write the liquid phase in ICON is in the temperature range from -10°C and one time from -15°C. Explain the difference between the two statements or be consistent.

- Figure 1: Increase size of Figure and or labels.

- Figure 2: Add in the figure caption that white space means no data availability.

- Figure 5: There is not dotted line in the figure (the ice line is dashed instead of dotted).

- Figure 7: The plot shows the period from June to July 2017- correct the caption accordingly.

- Figure 7: Use only one color for each category (now every color has another colorline at the top of the bar).

- Figure 9: Explain the $\Delta$ information in the x-axis labels in the caption.

- Figure 9: Is it not July 2017 instead of August 2017?

- Figure 9: Replace "Frequency" with "The frequency of occurence" in the caption.

- Figure 10: The x-axis labels are not formatted the right way (..=?)?

- Figure 10: What does the number mean on top of each column?

- Figure 11: Remove profiles in the figure labels for c and g (or add it in all other figures to be consistent).

- Table 1 and 2: Reformat the text to make the columns better readable.

- Table 1, 4th column, line 3: Add commas in between the different vertical resolutions.

- Table 2: Write each column capitalized at the beginning.

- Table 2: Add the temporal resolution, vertical resolution and retrieved parameters for radar, ceilometer and ICON.

---

## Author Comment (AC1) · 9 Feb 2019

Please find our detailed responses in the supplement

Please also note the supplement to this comment:
https://www.atmos-chem-phys-discuss.net/acp-2018-1144/acp-2018-1144-AC1-supplement.zip

---

## Author Comment (AC2) · 9 Feb 2019

Please find our detailed responses in the supplement

Please also note the supplement to this comment:
https://www.atmos-chem-phys-discuss.net/acp-2018-1144/acp-2018-1144-AC2-supplement.zip

---

## Author Response (AR1)

**Reply to referee#1**

We would like to thank the referee #1 for the constructive comments, which helped to improve the manuscript. We have considered all the recommendations. In the following reply we repeat the statements of the referee (in red) and the reply to each statement of the referee (in black). Note that in the statements of the referee the line and figure numbers refer to the original manuscript and may have changed in the revised version. In the replies the line and figure numbers correspond to the revised version.

P1,L16: clouds do not conserve LW radiation; "emit" is a better word here
- ✓ Corrected

P2,L3: midsummer SW radiative cooling of the surface is not a general feature across the Arctic. You might indicate that clouds cool the surface the entire year at a site like Summit Greenland (Miller et al. 2018)

- ✓ We added the following sentence (please, see the page 2, line 5):
  "This summer SW radiative cooling of the surface was reported for different Arctic regions except the Summit station in Greenland where the cloud radiative forcing effect is positive the entire year due to high surface albedo of the snow coverage (Miller et al. 2015, Miller et al. 2017)."

P2,L33: sensors cannot "capture low level clouds" –"observe" is a better word

- ✓ Corrected, (please, see the page 3, line 1).

P4,L29: "multi-variable"

- ✓ Corrected, (please, see the page 5, line 3).

P6,L27: I am not sure what these double values (e.g., "+100% / -30%") mean here. Please clarify

- ✓ The uncertainty here are given with respect to the reference IWC (from in-situ), i.e. for the ratio between retrieved and measured IWC. For instance, "+100% / -50%" means that the retrieved IWC is a factor of two larger or smaller than the measured IWC, respectively. The numbers here were taken from Hogan et al. (2006).
- ✓ There was a typo in the text regarding the uncertainty values. The numbers "+100% / -30%" were corrected to the right ones "+100% / -50%".
- ✓ Unfortunately, Hogan et al. (2006) do not provide numerical values for the temperature range from -40° to -20°C and it is hard to estimate the precise numbers from the figure they give. Therefore, we also check the uncertainties of the method in Heymsfield et al. 2007.
- ✓ We added the following sentence: "Hogan et al. 2006 found that uncertainties of the IWC retrieval differ for different temperature ranges and are estimated to be from -50% to +100% for temperatures below -40°C and ranging from -33% to +50% for temperatures above -20°C. The numbers here are root mean squared errors given with respect to the reference IWC. Evaluating the method of Hogan et al. 2006, Heymsfield et al. 2007 found similar uncertainties, except that there was a positive bias of about 50 % for temperatures above -30°C. The authors estimated the uncertainties from 0% to +100% and from -50% to +100% at temperatures above and below -30°C, respectively." (please, see the page 7, lines 12-17).

P7,L3: "US NationalWeather Service's"

- ✓ Corrected (please, see the page 7, line 27).

P10,L6: I found this minimum in July surprising. I suspect this is just a sampling error due to a relatively small dataset. I would encourage the authors to bring out this point more strongly here and in the conclusions of the paper

✓ The Cloudnet algorithm requires both radar and ceilometer measurements to be present, with the ceilometer able to distinguish cloud from warm precipitation. The ceilometer was out of operation for a few days in both July 2016 and July 2017, hence the reduction in Cloudnet data availability. We manually calculated the cloud fraction using only the radar measurements, and the cloud fraction results did not change much. Based on radar-only observations, which give a data availability of 99 %, we found the total cloud fraction to be around 80 %, with 19 % clear sky cases, very close to the Cloudnet (lidar and radar) calculations.

P11,L20-25: Shupe et al. 2015 also had the same challenges in distributing liquid water vertically in mixed-phase clouds. Does the CloudNet algorithm use the same scaled-adiabatic method? Perhaps mention both the Shupe paper and address this question here.

✓ Cloudnet also uses the scaled-adiabatic method for distributing liquid water vertically in liquid clouds (including liquid layers in mixed-phase clouds), using cloud boundaries from radar and lidar together with LWP from HATPRO. We added the reference to Shupe et al., (2015) which acknowledges the challenges of using this method in multi-layer situations. Please, note that in the paper we neither analyze LWC profiles nor focus on the detection of multiple liquid layers in mixed-phased clouds. We also do not try to classify multi-layer clouds into any further classes (like ice-ice, mixed-ice). In this paragraph we just explain why we do not take multi-layer cloud for further analysis. We describe the observational capabilities and limitations for liquid and mixed-phase multi-layer clouds.
✓ We modified the sentence (please, see the page 12, lines 13-15): "In situations with multiple liquid layers, whether warm or mixed-phase, partitioning the observed LWP from HATPRO among these different layers is particularly challenging and results in larger uncertainties (Shupe et al., 2015).".

P13,L1: There were not 26 NWP models in the Klein et al. study. Most were very experimental cloud-resolving or LES models.
✓ We removed the word "NWP" (please, see the page 13, line 33).

P14,L7: "represented" is a better word than "reprocessed"
✓ Corrected, (please, see the page 15, line 9).

P14,L10: What sort of radar reflectivity does this represent, given typical hydrometeor sizes? Perhaps a statement can be made here about the advantage of using an instrument simulator?
✓ We added the following sentence (please, see the page 15, lines 16-22): "According to the Z-IWC-T relation from Hogan et al (2006), the chosen threshold in the ice mixing ratio corresponds to the radar reflectivity factor ranging from -55 to -32 dBZ at temperatures from -60 °C to -5 °C. In general, these values are close to the radar sensitivity, although at high altitudes the radar sensitivity is about -40 dBZ (Küchler et al. 2017). Nevertheless, most of the observed hydrometeors are located within 2 km from the surface (see section 4.1) and, therefore, the lack of sensitivity at high altitudes does not significantly affect the results. For more detailed analysis of the uncertainties due to differences between the instrument and the model sensitivity can be done using observation simulators (e.g. Haynes et al. 2007). Such the analysis is out of the scope of the current study."

P14,L23: you state ": : :these ice-related processes are longer in the real cloud than it is specified in ICON model." Does this mean that you believe that the model has the processes act more rapidly than they do in nature? This sentence needs clarification
✓ As we show the highest occurrence of single-layer ice clouds in the ICON model is at relative humidity over ice of up to 110 %. Real clouds are observed at higher ice supersaturation

conditions (125 %). This might be associated with a different number concentration of ice particles or/and depositional growth efficiency. For instance, if the number concentration of ice particles parameterized in the ICON model is higher than the real one, the water vapour deposits faster onto the ice particles due to larger total surface area of particles and the conditions of higher supersaturation (125 %) might be not reached. The number of formed particles depends on the nucleation processes. Therefore, the difference in the concentration might come from the parameterization of ice nuclei in the ICON model.

✓ The sentences were modified (please, see the page 15, lines 29-32, page 16, lines 1-2): "The presence of ice particles at lower supersaturation over ice in the ICON model in comparison with observations may be associated with ice nuclei (IN) parameterization in the ICON model, which is known to be still a challenge (Fu et al., 2017). We speculate that a higher concentration of IN and, thus ice particles, leads to faster deposition of water vapour onto the ice particle's surface. Therefore, a more efficient vapour-to-ice transition in the model could lead to lower relative humidity. Similarly, the parameterization of deposition growth rate and secondary ice processes may also have an impact on the in-cloud relative humidity."

P15,L19: "categorization"→"classification"

✓ Corrected, (please, see the page 16, line 30).

P16,L33:These clouds with LWP values between 30 and 60 g/m2 also have the largest radiative contribution to the surface energy budget (Bennartz et al. 2013)

✓ Added, (please, see the page 18, lines 11-12).

P18,L3: The L&C paper only talk about higher frequency microwave radiometer measurements. If you want to include the IR here (and you should), then please reference either Marke et al. 2016 or Turner 2007

✓ Corrected. Both references were included, (please, see the page 19, lines 12-13).

P18,L6: how would the effective radius be determined?

✓ Modified, (please, see the page 19, lines 16-17): "The next step will be to derive cloud microphysical properties such as LWC, IWC and effective radius for different types of clouds using methods by Frisch et al. 1998, Frisch et al. 2002, Hogan et al. 2006, Delanoe et al. 2007."

Fig 9, panels B and C: the most important clouds from a radiative point of view are the ones with smaller LWP and IWP. However, the linear x-axis sort of hides them. Would you consider plotting these data on a log10 axis from 1 g/m2 to 1000 g/m2 for LWP? The bins should be equal-sized bins in log10(LWP) space. Ditto for the IWP, but perhaps start the lower range at 0.1 g/m2?

✓ Please find the plot below this answer. As mentioned in the section 2.2 the uncertainty of the LWP retrieval based on the MWR observations is 20-25 g/m^2. Thus, we do not see a need to subdivide the first bin into smaller ones. We also think that it could be a bit confusing for a reader to look at the logarithmic distributions of LWP and IWP. The bins have significantly different width in the linear domain and cannot be compared with each other. This also lead to significant changes of the distribution peaks which may be misleading.

[Figure]

Fig11: I don't see the labels for the individual panels (e.g., "a", "b", etc). Also, there seems to be some weird inset artifact on the lower left-hand panel.
- ✓ Corrected, (please, see the Fig.13).
- ✓ The font size of labels was changed.

Fig12: I would emphasize that the ICON results are not "global", but are from output over the Ny Alesund

- ✓ Corrected, (please, see the captions to the Fig.13 and Fig.14).

**Reply to referee#2**

We would like to thank the referee #2 for the useful and constructive comments, which helped us significantly improve the manuscript. We have considered all the recommendations. In the following reply we repeat the statements of referee (in red) and the reply on each statement of referee (in black). Note that in the statements of the referee the line and figure numbers refer to the original manuscript and may have changed in the revised version. In the replies the line and figure numbers correspond to the revised version.

**1 Major comments:**

- The cloudnet algorithm seams to use thermodynamic profiles of a NWP model. Please explain what that exactly means and how that influences the analysis? If the thermodynamic profiles deviate a lot from real conditions (as measured by radiosounding for example) what are the consequences? And could that probably lead to inconsistency when using in a further analysis later on radiosoundings as additional information?
    - ✓ The Cloudnet algorithm uses the vertical profile of temperature, pressure and humidity from a forecast model (e.g. ECMWF or ICON). The temperature profile is used to aid determination of the freezing level, which is then refined in precipitating cases using the step change in the observed vertical velocity. The temperature profile is also used in the derivation of IWC, which uses reflectivity and temperature as inputs. Uncertainty in IWC is obtained by propagating the expected uncertainties in radar reflectivity and the model temperature. The forecast models used will have assimilated radiosoundings if available and are typically within 1.5 °C. Note that a radiosounding profile is not collocated with respect to the radar and lidar observation column as it can drift more than 100 km horizontally by the time it reaches the tropopause.
    - ✓ Thermodynamic profiles are used for an identification of the melting layer (0 °C isotherm). The model uncertainties may lead to the liquid-ice misclassification at temperatures close to 0 °C. The difference of temperature and relative humidity profiles of the column output of the NWP ICON model over Ny-Ålesund and the radiosonde data is shown in Fig.a1. Around 300 radiosonde profiles were included. The comparison reveals that up to 10 km the maximum errors in temperature and relative humidity are -1.5+/-1.5 °C and -5+/-20 %, respectively.

[Figure]

Fig. a1 Difference in temperature (a) and relative humidity (b) between ICON column output over Ny-Ålesund and radiosonde data. Blue and red lines show the bias and the standard deviation, respectively.

In the case of precipitating clouds uncertainties of the model are further mitigated by the Cloudnet algorithm using radar Doppler observations. The algorithm looks for a layer with a significant gradient in the particle vertical velocity. This layer is then classified as melted particles. Below this layer particles are classified as liquid hydrometeors, above depending on what the lidar observes particles are classified as ice, supercooled liquid, or both. For the classification the Cloudnet algorithm identifies the 0 °C isotherm using the wet-bulb temperature calculated from the model data. Therefore, the model uncertainties (Figs.1 and 2) may lead to the liquid-ice misclassification at temperatures close to 0°C. In the case of precipitating clouds uncertainties of the model are mitigated by the Cloudnet algorithm using radar Doppler observations. The algorithm identifies the 0°C isotherm by a significant gradient in the particles vertical velocity.

✓ We have added the information in Sec. 2.5: "For the classification the Cloudnet algorithm identifies the 0 °C isotherm using the wet-bulb temperature calculated from the model data. Therefore, the model uncertainties (Figs.1 and 2) may lead to the liquid-ice misclassification at temperatures close to 0 °C. In the case of precipitating clouds uncertainties of the model are mitigated by the Cloudnet algorithm using radar Doppler observations. The algorithm identifies the 0 °C isotherm by a significant gradient in the particles vertical velocity.", (please see the page 7, lines 5-8).

✓ The model output is used for the correction of the gas/liquid absorption of the radar signal. The two-way uncertainty of the gas-attenuation estimated by Hogan and O'Connor (2004) is about 10% (added to sec 2.4). In this particular study this correction may only influence IWP retrievals. Nevertheless, the introduced uncertainties are much smaller than the uncertainties of the IWP retrieval itself.

- ✓ We added the sentence, (please see page 6, lines 12-14): "The Cloudnet algorithm corrects the radar reflectivity for the attenuation by atmospheric gases and liquid water. Temperature, humidity, and pressure profiles from a model are used by the Cloudnet for the corrections. The two-way uncertainty of the gas-attenuation estimated by Hogan and O'Connor (2004) is about 10%.".

- Discuss more the uncertainties of the measurements and their role in terms of interpretation of the results, e.g. the uncertainty range of IWC (which seems to be quite large). This could be of importance especially when it comes to comparisons. Include this discussion also in the conclusions when discussing the comparison of observations with ICON model output.

  - ✓ "The ceilometer is calibrated using the technique by O'Connor et al (2004), which has uncertainties of 10%" (this has been added to sec 2.3). Within this study the ceilometer observations are used for the liquid layer detection. Lidars are very sensitive to liquid particles and therefore the observation uncertainty does not significantly influences the results. Nevertheless, as we mentioned in Sec. 2.3 the optical signal experiences a strong attenuation and, therefore, the ceilometer cannot reliably detect multiple liquid layers. This makes the classification of multiple layer clouds difficult, therefore these clouds are excluded from the analysis.

  - ✓ As discussed in the Sec. 2.2, LWP measured by the MWR has uncertainties of 25 g/m^2. LWP is not used for the cloud classification. It is used for the correction of the radar reflectivity for the liquid water attenuation. The uncertainty of 25 g/m^2 causes about +/- 0.2 dB uncertainty in the two-way attenuation at the W-band (Matrosov et al. 2009). The following abstract was added to Sec. 2.4: "The Cloudnet algorithm corrects the radar reflectivity for the attenuation by atmospheric gases and liquid water. Temperature, humidity, and pressure profiles from a model are used by the Cloudnet for the corrections. The two-way uncertainty of the gas-attenuation estimated by (Hogan et al. 2004) is about 10%. The uncertainty of 25 g/m^2 in LWP from MWR causes about +/-0.2 dB uncertainty in the two-way attenuation at the W-band (Matrosov et al. 2009)."

  - ✓ We added the information on the radar reflectivity uncertainty to Sec. 2.4: "The total radar reflectivity uncertainty consists of the calibration bias which is within +/- 0.5 dB (Kuechler 2017), the random error, and the gas/liquid attenuation uncertainty. The random error depends on a number of independent measurements, which is for the 30 s Cloudnet sampling varies from 72 to 108 for the used radar settings (see Table 1). Taking into account the non-coherent averaging of the independent measurements (Bringi and Chandrasekar 2001, Eq 5.193) the standard deviation of the random error is in the order of 0.5 dB."

  - ✓ The uncertainty in the radar reflectivity only influences IWC retrieval. The total uncertainty of 2 dB corresponds to about +40/-30% uncertainty. Part of this uncertainty is likely to be included into the uncertainty of the Z-IWC-T relation from Hogan et al (2006) because the relation was found empirically using radar observations. For the IWP calculation a number of range bins is integrated, which reduces the influence of the random error. We added the following abstract to Sec. 2.5: "The uncertainty in the radar reflectivity also influences IWC retrieval. The total uncertainty of 2 dB corresponds to about +40/-30% uncertainty. Part of this uncertainty is likely to be included into the uncertainty of the Z-IWC-T relation from Hogan et al (2006) because the relation was found empirically using radar observations."

  - ✓ Please note, that the uncertainties mentioned above only affect quantitative retrievals such as IWC. These uncertainties do not affect the classification results. Only the cloud classification from observations and the model was compared. Therefore, the uncertainties do not change the results of comparison.

- In section 5 cloudnet products are used in combination with radiosoundings in comparison to ICON model output. I am missing one essential discussion here: How different are the temperature and humidity profiles from ICON compared to the radiosoundings? If the thermodynamic profiles from ICON deviate considerably from the data from radiosoundings the analysis could be limited by that independent of the cloud information derived by cloudnet or from ICON.

✓ We compared temperature and relative humidity profiles of the radiosondes data and column output of the NWP ICON model over Ny-Ålesund (please see answer to the first question). The comparison showed that up to 10 km the maximum errors in temperature and relative humidity are -1.5+/-1.5 °C and -5+/-20 %, respectively. The width of bins in temperature and relative humidity in Figs. 11 and 12 are 5 °C and 10 %, respectively. Since the biases of temperature and relative humidity are smaller than width of bins and, therefore, these biases do no change the results of model and observation comparison. For the comparison we use more than 300 radiosondes, therefore possible effects of the random errors are mitigated.

✓ The figures added to the Sec. 2.6. We added the sentences in Sec.2.6.1: "The uncertainties of the temperature and relative humidity profiles of GDAS1 are shown in Fig.1. The maximum errors in temperature and relative humidity do not exceed -1+/-1.5°C and -15+/-24%, respectively." The following sentences were added in Sec.2.6.2: "The uncertainties of the temperature and relative humidity profiles of the ICON model are shown in Fig.2. The maximum errors in temperature and relative humidity at the altitude up to 10~km are -1.5+/-1.5°C and -5+/-20%, respectively.".

- Discuss more what the thermodynamic conditions mean when related to cloud properties. What can be derived from this information and what is still missing? Does not for cloud formation also the stability or layer of the atmosphere (boundary layer, inversion etc.) play a huge role? Could that be analysed with the used dataset as well?

  ✓ A number of studies show a dependence of particle formation processes on the cloud top temperature. For instance, laboratory studies show that shapes of ice crystals are defined by the environment temperature and humidity. There is also some evidence that similar effects happen in the real atmosphere. Aggregation efficiency and deposition growth rate are also temperature and humidity dependent. In order to improve the parameterization of these processes in models, further studies on particles formation and microphysical properties are needed. We added the following sentences to Sec 4.3: "For instance, laboratory studies show that shapes of ice crystals are defined by the environment temperature and humidity (Fukuta and Takahashi, 1999, Bailey and Hallett, 2009). There is also some evidence that similar effects happen in the real atmosphere (Hogan et al., 2002, Hogan et al., 2003b, Myagkov et al., 2016). Aggregation efficiency and deposition growth rate are also temperature and humidity dependent (Hosler and Hallgren, 1960; Bailey and Hallett, 2004, Connolly et al., 2012)."

  ✓ In this study we have shown that the model output and the remote observations of clouds show significant differences. In sec. 5 we speculate about the possible reasons, although further studies are needed in order to find an answer. We rephrase the following sentences in Sec.5: "Both, the ICON model and observations, reveal that ice particles typically occur at relative humidities higher than the saturation over ice but lower than the saturation over water. The high occurrence of ice phase in ICON is found at RHi up to 110%, while for the observations reveal RHi of up to 125% (Fig.13b). The presence of ice particles at lower supersaturation over ice in the ICON model in comparison with observations may be associated with ice nuclei (IN) parameterization in the ICON model, which is known to be still a challenge (Fu and Xue, 2017). We speculate that a higher concentration of IN and, thus ice particles, leads to faster deposition of water vapour onto the ice particle's surface. Therefore, a more efficient vapour-to-ice transition in the model could lead to lower relative humidity. Similarly, the parameterization of deposition growth rate and secondary ice processes may also have an impact on the in-cloud relative humidity.".

  ✓ The aerosol concentration and chemical composition also significantly affect the particle formation since not all of them can act as ice nuclei and cloud condensation nuclei. Additional studies using Raman lidar measurements and in-situ observations of aerosols might be useful to understand the nucleation processes. The atmospheric stability and occurrence of inversions also play a huge role in cloud formation. Nevertheless, models often miss inversions, and therefore, we do not analyze them in this study.

✓ Within this study we focus on the occurrence of clouds at Ny-Ålesund and try to show at which environmental conditions they occur. We found the relation between the different types of clouds to in-cloud conditions (temperature and humidity). This information is relevant for our further cloud radiative studies. In addition, the information given in this paper can be compared to what have been found at other Arctic sites to see the difference among them.

**2 Specific comments:**

- Add the vertical range of each instrument (lowest detection, maximum profilig range). Especially the lowest detection could be interesting to know.
    - ✓ The lowest detection for the FMCW 94 GHz cloud radar is 100 m and the maximum is 10 km (the information can be found in Table 1). For ceilometer CL 51 the minimum range is 10 m, the maximum range 15 km.
- Page 6, line 27: What is the IWC uncertainty range between -40 and -20_? Please add that information as well.
    - ✓ The numbers here were taken from Hogan et al. (2006). Hogan et al. 2006 found that uncertainties of the IWC retrieval differ for different temperature ranges and are estimated to be from -50% to +100% for temperatures below -40°C and ranging from -33% to +50% for temperatures above -20°C. The numbers here are root mean squared errors given with respect to the reference IWC.
    - ✓ Unfortunately, Hogan et al. (2006) do not provide numerical values for the temperature range from -40° to -20°C and it is hard to estimate the precise numbers from the figure they give. This method was evaluated in Heymsfield et al. 2007, who found similar who found similar uncertainties, except that there was a positive bias of about 50 % for temperatures above -30°C. The uncertainties for the temperature range from -40° to -20°C appear to be slightly larger than for temperatures above -20°C, but lower than for temperatures below -40°C, hence we will assume the large of these two (i.e. +100%/-50%).
    - ✓ We added the following sentence: "Hogan et al. 2006 found that uncertainties of the IWC retrieval differ for different temperature ranges and are estimated to be from -50% to +100% for temperatures below -40°C and ranging from -33% to +50% for temperatures above -20°C. The numbers here are root mean squared errors given with respect to the reference IWC. Evaluating the method of Hogan et al. 2006, Heymsfield et al. 2007 found similar uncertainties, except that there was a positive bias of about 50 % for temperatures above -30°C. The authors estimated the uncertainties from 0% to +100% and from -50% to +100% at temperatures above and below -30°C, respectively."
- Page 7, line 5: Is it not a 6 hour dataset if the data is generated at 00, 06... ? Please correct or explain.
    - ✓ The temporal resolution of GDAS1 dataset is 3 hours.
    - ✓ The sentences in Sec. 2.6.1: "The GDAS data are generated at 00, 06, 12 and 18 UTC. The program produces a 3 hour dataset on 23 pressure levels." were modified to "The GDAS dataset is initialised every 6 hours and outputs an analysis timestep followed by forecasts with a temporal resolution of 3 hours."
- Page 7, line 19: Why was the ICON output not used consistent for the whole time period of analysis? What is the difference introduced by using ICON compared to GDAS1? Which uncertainty results from using one or the other? Explain more what the influence is of the model data used on the cloudnet products and thus on your analysis.
    - ✓ The ICON model data only became available from the 1st February 2017, when the ICON column output for Ny-Ålesund was added to the extraction process. GDAS1 model data was used prior to this. ICON model data has the advantage of higher vertical, horizontal and temporal resolution. ICON data has a temporal resolution of 1 hour rather than 3 hours for GDAS1. The vertical resolution of ICON column output ranges from 30 m at the lowest height, 255 m at 2 km and up to 400 m at 10 km. The

vertical resolution for GDAS1 is coarser: 233 m at the lowest altitudes, 500 m at 2 km and 1200 m at 10 km. The comparison of ICON and GDAS1 temperature profiles with radiosonde (Fig.a1 in the answer on the first question and Fig.a2) showed that at the lowest 1 km the GDAS1 temperature bias (0.5 °C) is lower than for the ICON model column output (<1.5 °C), but for the highest altitudes up to 10 km ICON has lower bias (0.5 °C) in comparison with GDAS1 (1 °C). The comparison of relative humidity profiles in respect to radiosonde data showed that the bias for both ICON model and GDAS1 is not exceeding 5 % with a standard deviation of 20 % except for the lowest 1 km where the bias for ICON (<5 %) is lower than for GDAS (<15 %). Since the Cloudnet algorithm uses the temperature and humidity profiles as an input it might affect phase partitioning. Nevertheless, the differences between GDAS1 and ICON are not large. In comparison with the previously used GDAS1 data the ICON model column output has a better temporal and vertical range resolution. The biases of temperature and humidity for the ICON model output are in general lower.

[Figure]

Fig. a2 Difference in temperature (a) and relative humidity (b) between GDAS1 and radiosonde data. Blue and red lines show the bias and the standard deviation, respectively.

- Page 7, line 24: Why was a forecast product used and not nudged simulations using reanalysis data? Would it not be more correct to use nudged simulations or are there technical limitations? Is the vertical profile data the same as used for Cloudnet?
  - ✓ Forecast products are preferred when using model data in Cloudnet since the ultimate goal is evaluation of the model representation of clouds.
    Clouds are challenging for current data assimilation methods, with any cloud variables in analysis fields rapidly changing in the first few hours as a forecast run progresses - the forecast model 'spin-up' problem.
    Additionally, analysis fields with 'full' assimilation are usually only available at 00 and 12Z, and the thermodynamic profile may change significantly within this time - forecast output is usually available with a temporal resolution of 1 hour.
- Page 9, line 4: How was the monthly mean IWV estimated? From radiosonde profiles?

- ✓ Yes, the IWV was derived from radiosonde data. The IWV for our period of study was compared with one for previous 23 years defined by the radiosonde availability. The information is given in Sec.2.1 and Table 1.

- Section 3: What is the conclusion from your analysis about the thermodynamic conditions? How much can the results be generalized or what does it mean for the findings of the paper that the year 2017 was different from previous years?
  - ✓ In the Section 3 we have characterized the thermodynamic conditions for our period of study since the temperature and IWV are one of the important parameters that define cloud formation and development. We have compared temperature and IWV to previous years to show how the atmospheric conditions of our period of study differ from previous years. And some months of our study have specific features. We think this is very important for proper comparisons of the results from different time periods. A good example is the 7[th] comment of the first reviewer, who was surprised by the low occurrence of clouds in July 2016. The reviewer expected a higher frequency based on previous observations. One of the reasons could be different thermodynamic conditions. Therefore, we show how temperature and IWV differ from earlier observations. Another example would be October 2016 has the highest cloud occurrence and this was associated with the warm temperature anomaly. Thus, some of the months of our period of study have high or low cloud occurrence and it might be due to the specific thermodynamic conditions.
  - ✓ In order to understand how exactly thermodynamic conditions influence the cloud occurrence in Ny-Ålesund a multi-year dataset is required. We plan to focus on this topic in the future.

- Page 11, line 29: Please explain how the Cloudnet IWC product is estimated.
  - ✓ The Cloudnet IWC is derived from the radar reflectivity factor and the temperature at the altitude analyzed (Hogan et al. 2006; Heymsfield et al., 2008). We referenced these sources in the Section 2 (page 7, lines 9-11).

- Page 11, line 33/Section 4.3: How does not taking into account cases with liquid precipitation influence the results due to bias for specific cloud types? Or is there no cloud type bias for liquid precipitation?
  - ✓ We focus on single-layer clouds without liquid precipitation since the lidar signal is significantly attenuated in the liquid precipitation, and thus, leading to underestimation of upper liquid layers. Moreover, LWP is not available when the HATPRO radome is wet.
  - ✓ We added the sentence "Thus, all results are relevant for single-layer clouds without liquid precipitation." (please, see page 12, line 28).

- Page 12, line 8: How is the geometrical thickness of the single-layer clouds estimated?
  - ✓ The geometrical thickness of a cloud is calculated as the cloud top altitude minus cloud bottom altitude. The cloud top/bottom altitude is the upper/lower border of the uppermost/lowermost range bin with the cloud. This is added to Sec. 4.3.

- Page 12, line 27: Specify the model domain/region of the 6 regional models.
  - ✓ The information on the region was added, and the sentence in the Sec. 4.3 was changed to "Tjernstrom et al. (2008) evaluated 6 regional models which were set to a common domain over the western Arctic and found that…"

- Page 13, line 1: Which region/domain was the comparison focused on?
  - ✓ The information was added, and the sentence was replaced by "Klein et al. (2009) compared 26 models with air-borne and ground-based observations over the north Alaska (Barrow and Oliktok Point).".

- Page 13, line 22: How would correcting for the drift influence the results or how much uncertainty is introduced by not doing so?
  - ✓ According to McGrath et al (2006) the uncertainties in temperature due to the radiosonde drifts in the northern hemisphere do not exceed 0.4 deg C up to 10 km altitude. Uncertainties in relative humidity are about 3 %. The uncertainties are small compared to the uncertainties of the model and do not significantly change the results.

- ✓ We added the sentences to the Sec.4.3: "According to McGrath et al (2006) the uncertainties in temperature due to the radiosonde drifts in the northern hemisphere do not exceed 0.4 °C up to 10 km altitude. Uncertainties in relative humidity are about 3 %."
- Page 14, line 20: Explain how the ice nuclei information has to be specified in ICON.
  - ✓ Please note that in this paper we just show that the appearance of clouds in the observations and in the model is different. We speculate about possible reasons of the difference, but the exact reason is still not known. We make no recommendations regarding the parameterizations of processes in the ICON model.
- Page 14, line 21: Why is the IN concentration increasing? Explain your argumentation here better.
  - ✓ We agree the formulation is misleading. The sentences were modified: "The presence of ice particles at lower supersaturation over ice in the ICON model in comparison with observations may be associated with ice nuclei (IN) parameterization in the ICON model, which is known to be still a challenge (Fu et al., 2017). We speculate that a higher concentration of IN and, thus ice particles, leads to faster deposition of water vapour onto the ice particle's surface. Therefore, a more efficient vapour-to-ice transition in the model could lead to lower relative humidity. Similarly, the parameterization of deposition growth rate and secondary ice processes may also have an impact on the in-cloud relative humidity."
- Section 5: Discuss how the model resolution of ICON could influence the results and your interpretation.
  - ✓ The ICON model vertical and horizontal resolution would influence the results. Most of the clouds occur at the height around 2 km where the vertical resolution of ICON model is about 260 m. More studies are needed to investigate the influence of the model resolution by changing its vertical resolution and looking to the affect. This is out of the scope of the current study. The intention of our paper is not the direct evaluation of the ICON model but rather to show an application of cloud observations for a model evaluation.
- Page 17, line 9: You write that single-layer ice clouds are thicker than mixed-phase clouds. Why is that? Do cloud droplets not increase the cloud thickness?
  - ✓ Please note that often ice particles in mixed-phase clouds are large and reach the 0 °C isotherm (especially in the cases of deep convection). These particles melt and form liquid precipitation. These cases, which are typically thick, are excluded from the analysis. The mixed-phase clouds with no liquid precipitation seem to be thinner.
- Figure 6: It would be interesting to add another subplot of the subfigures b and c showing the profiles for the lower 2 km.
  - ✓ All the main features can be seen from these plots and we think that additional subplots are not needed.
- Figure 8: Why does the liquid precipitation only occur in mixed-phase clouds? Are there no liquid clouds with liquid precipitation?
  - ✓ Liquid precipitation also occurs in liquid clouds. In the Fig.8 two lines for mixed-phase clouds are plotted to show the difference in seasonal variability of mixed-phase clouds with liquid precipitation and without.
  - ✓ We added the line for single-layer liquid clouds with liquid precipitation in figure 8.
  - ✓ Figure 8 shows that the occurrence of single-layer liquid clouds without liquid precipitation is slightly different from the single-layer liquid clouds with liquid precipitation. For the single-layer mixed-phase clouds the difference is larger.
  - ✓ We added the sentence: "This is in agreement with Mülmenstädt et al., 2015 who reported that most of the liquid precipitation is formed including the ice phase.".
- Figure 11: How are single-layer clouds defined in ICON? As one model level cloud free in between model level containing clouds? Please specify.
  - ✓ As mentioned in Sec. 5 the ICON-based classification is made in the same way as for Cloudnet (please see Sec. 4.2). We detected clouds in ICON model output using the threshold of $10^{-7}$ kg/kg in the

ice/liquid mixing ratio. Each single-layer cloud profile was defined as a continuous set of range bins without any cloud-free areas in between.

- Figure 11: What is the time resolution of the ICON output used for the analysis compared to the observations? How would the observations look like if not the time period of one hour around the radiosounding would be used but instead the time period of the ICON output? How does the model resolution thus influence the results?
  - ✓ The time resolution of ICON model output is 1 hour (section 2.6.2). And we look at the same time period (+/- 1 hour from radiosonde launch) in model and observations. Since we look at a long time period of several month the statistic is still robust. Thus, the ICON model temporal resolution should not strongly influence the results. In addition, in this study we focus on the conditions in general under which clouds occur and not a direct model evaluation.
- Figure 11 and 12: You mention here that the global ICON model is used- how different is the resolution compared to the single-column version mentioned earlier? Which domain size/how many grid cells were analysed?
  - ✓ For the analysis we used the mentioned earlier column output for the grid box where Ny-Ålesund is located in and we changed the labels to figures 13 and 14.
  - ✓ In the label to figure 13 we replaced the phrase "…for global NWP model ICON (right)." to "… for column output of NWP model ICON over Ny-Ålesund (right)."
  - ✓ In the label to figure 14 we replaced the phrase "…for global NWP ICON model (b)." to "… for column output of NWP ICON model over Ny-Ålesund (b)."
- Figure 12: There is considerably less profiles analysed from the ICON model compared to the observations- how does that influence the analysis? Is the statistics sufficient for general statements about the clouds?
  - ✓ Please note, that in Fig. 12 (Fig.14 in revised version) N is not a number of profiles but a number of pixels (range-time bins). Since Cloudnet has significantly higher range resolution than ICON it has larger values of N. We analyzed a time period of several months (same for observations and the model) and got the results consistent with other studies (please see Sec.5). A larger dataset would be of course beneficial for more detailed analysis.

**3 Small remarks, typos:**

- Write out abbreviations in the abstract (AWIPEV, LWP, IWP, ICON).
  - ✓ Corrected
- The numbering of the subsections is not always correct, e.g. p. 5, l. 12 and p. 5, l. 27 subsection 2.2.5 has to be replaced by subsection 2.5, p.12 l. 20 Sect. 22.2 does not exist, p. 13 l. 19 Sec. 4.4.2 does not exist.
  - ✓ Corrected
- The analysed period is not mentioned consistent throughout the manuscript. From the plots etc. it looks like as the analysed period is from middle of June to July 2017. However in the manuscript it is often stated that 14 months are analysed. Also at some place the period is mentioned with different start/end dates, e.g. p. 6 l. 29: August, p. 7 l. 9: 31 January 2017.
  - ✓ Page 7, line 21: "August" was corrected to "July".
  - ✓ On page 7, line 9 "31 January 2017" is correct since this date is related to the end of using the GDAS1 data as an input for Cloudnet algorithm. Then since the February 1 2017 the ICON model column output was used.
  - ✓ Page 16, line 29: The word "almost" was added to the sentence since the data was available since 10th of June. "We analyzed almost a 14-month…"
- Leave away the article in front of ICON or add model (in ICON … or in the ICON modell…).
  - ✓ Page 8, line 15: the word "model" was added.
  - ✓ The article "the" was added before "ICON model" on page 15, line 28,31 and 32.
- Some figures (e.g. Fig. 5) seam not to be vectorized and thus have not such a good resolution when zooming in- change if possible.

- ✓ For us everything looks fine with figure 5 (in revised version Fig.7). Also, the pdf file of figure 5 (in revised version Fig.7) looks fine and is vectorized. Maybe, this technical issue will be checked again in the end.
- Page 1, line 10: delete the (to ICON model output).
  - ✓ Corrected
- Page 1, line 11-13: Rewrite the sentence/split it into two (Distinct ...evident.).
  - ✓ Corrected
- Page 1, line 15: Add \budget" (in the energy budget).
  - ✓ Corrected
  - ✓ The word "budget" was added after "energy".
- Page 4, line 6: Add model data, which is also included in Table 2.
  - ✓ Corrected
  - ✓ We made a separate sentence and added the information in the end of this sentence: "Table 2 gives an overview on the Cloudnet products used for the cloud analysis and provides input parameters and model data for the Cloudnet algorithm."
- Page 4, line 9: Why 21st of May? The analysis starts in June 2006.
  - ✓ Please note that the 21st of May 2006 to 2 May 2017 are in general the period which corresponds to the data availability of radiosonde (type Vaisala RS92). The cloud analysis starts in June 2016. Later we mentioned the period of radiosonde data that we used for our study (please see page 4, line 15).
- Page 4, line 10: Which radiosonde type was used after 2 May 2017?
  - ✓ Since 2 May 2017 the type of radiosonde was changed to the Vaisala RS41.
  - ✓ The following sentences were added on page 4, line 18: "Since 2 May 2017 the type of radiosonde was changed to the new radiosonde type Vaisala RS41. The accuracy of the RS41 radiosonde type reported by manufacturer is 0.1°C for temperature and <2% for humidity.".
- Page 5, line 25: There is a \." too much in front of the citation.
  - ✓ Corrected
- Page 6, line 27 and page 15, line 13: Add unit after temperature (C).
  - ✓ Corrected
- Page 7, line 23: Replace compared with compare.
  - ✓ Corrected
- Page 8, line 5: Do you mean less variable instead of milder?
  - ✓ Corrected, changed to "less variable".
- Page 8, line 33: Replace K with _C to be consistent.
  - ✓ Corrected
- Page 11, line 26: Change header to \Single-layer clouds and their relation to thermodynamic consitions".
  - ✓ Corrected
- Page 11, line 31f: Add percentages for all numbers.
  - ✓ Corrected
  - ✓ The percentages for all numbers of profiles were added in brackets.
- Page 14, line 1: Replace Figure11d with Fig. 11d.
  - ✓ Corrected
- Page 16, line 3: Replace \the different time period" with \a different time period".
  - ✓ Corrected
- Page 17, line 12: Replace \cloud models" with NWP or climate models (depending on what you want to say here).
  - ✓ Corrected
  - ✓ "cloud models" replaced by "NWP models".
- Page 17, line 20 and 22: One time you write the liquid phase in ICON is in the temperature range from -10_C and one time from -15_C. Explain the difference between the two statements or be consistent.

- ✓ On page 17, line 20 we write about the liquid phase in ICON which occur in the temperature range from -10 ° to +5°C. Line 22 is about single-layer mixed-phase clouds which occur from -15° to +0°C. Please note that we define mixed-phase clouds as a continuos cloud layer in a vertical profile with ice and liquid phases detected within the cloud boundaries. For example, with a cloud top with -25°C and liquid water layer detected only at -5°C the cloud will be considered as mixed-phase cloud. Therefore, mixed-phase clouds in general may occur at lower temperatures than liquid phase.
  - ✓ The definition of mixed-phase clouds is given in section 4.2.
- Figure 1: Increase size of Figure and or labels.
  - ✓ Corrected
  - ✓ The size of figure was adjusted to the two-column width.
- Figure 2: Add in the figure caption that white space means no data availability.
  - ✓ Corrected
  - ✓ The phrase "…, white space means no Cloudnet data availability." was added to the figure caption.
- Figure 5: There is not dotted line in the figure (the ice line is dashed instead of dotted).
  - ✓ Corrected
  - ✓ Page 9, line 32 "dotted black line" was replaced by "densely dashed black line".
- Figure 7: The plot shows the period from June to July 2017- correct the caption accordingly.
  - ✓ Corrected
- Figure 7: Use only one color for each category (now every color has another colorline at the top of the bar).
  - ✓ This comment is not clear for us. Everything looks fine in our pdf version. Also, the pdf file of figure 7 looks fine for us. Maybe, this is a technical issue and this will be checked again in the end.
- Figure 9: Explain the Δ information in the x-axis labels in the caption.
  - ✓ Corrected
  - ✓ We added the sentence in the caption to figure 9: "In x-axes Δ shows the bin width."
  - ✓ Current version of caption: "Frequency of occurrence of cloud thickness for single-layer clouds (a), of the logarithm of LWP for single-layer liquid and mixed-phase clouds (b) and of IWP for single-layer ice and mixed-phase clouds (c) for the period from June 2016 to July 2017. The y-axis is shown in logarithmic scale. In x-axes Δ shows the bin width. The frequency of occurrence is normalized by the total number of corresponding cloud type."
- Figure 9: Is it not July 2017 instead of August 2017?
  - ✓ Corrected
  - ✓ The caption to figure 9 was corrected, "August 2017" was replaced by "July 2017".
- Figure 9: Replace \Frequency" with \The frequency of occurence" in the caption.
  - ✓ Corrected
- Figure 10: The x-axis labels are not formatted the right way (..=?)?
  - ✓ Corrected
  - ✓ We changed the labels of x-axis.
- Figure 10: What does the number mean on top of each column?
  - ✓ Numbers at the top of plot show the number of cases included in the temperature range.
  - ✓ We mentioned the explanation in the caption to Fig.10.
- Figure 11: Remove profiles in the figure labels for c and g (or add it in all other figures to be consistent).
  - ✓ Corrected
- Table 1 and 2: Reformat the text to make the columns better readable.
  - ✓ Corrected
- Table 1, 4th column, line 3: Add commas in between the different vertical resolutions.
  - ✓ Corrected
- Table 2: Write each column capitalized at the beginning.
  - ✓ Corrected

- Table 2: Add the temporal resolution, vertical resolution and retrieved parameters for radar, ceilometer and ICON.
    - ✓ The temporal and vertical resolution and retrieved parameters mentioned in Table 2 were given for Cloudnet product.
    - ✓ We change the format of the Table 2.

**Additional changes**

Note that the line numbers refer to the revised manuscript.

1) **Page 4, Line 17:** The units of temperature were replaced from " K" to "°C" after 0.25 and 0.15.
2) **Page 7, Lines 10-11:** We corrected an article and the name of parameter. "… where Z is a radar reflectivity factor and T is a temperature."  Was replaced by "… where Z is the radar reflectivity factor and T the air temperature."
3) **Page 7, Line 12:** There was a typo in the text of the uncertainty values. The numbers "+100% / -30%" were corrected to the right one "+100% / -50%". Instead of using the symbol "/" we provide the uncertainty range. The units of temperature ranges were corrected by adding the symbol of degree Celsius and the reference to Horgan et al. (2006) was added.
   The final version of the sentence is: "Hogan et al. 2006 found that uncertainties of the IWC retrieval differ for different temperature ranges and are estimated to be from -50% to +100% for temperatures below -40°C and ranging from -33% to +50% for temperatures above -20°C. The numbers here are root mean squared errors given with respect to the reference IWC. Evaluating the method of Hogan et al. 2006, Heymsfield et al. 2007 estimated the uncertainties of +100% and from -50% to +100% in the temperature above and below -30°C, respectively.".
4) **Page 7, Line 21:** We removed the repeated word "availability" placed after "… more than 90 %".
5) **Pages 7-8, Lines 33-34:** We added the information on the GDAS1 vertical resolution at 2 km height to the sentence: "The vertical resolution varies from 173 m near the ground to 500 m at the height 2 km and to ~2.5 km at the height of 15 km."
6) **Page 8, Line 9-10:** We added the sentence about the ICON vertical resolution at 2 km height: "The vertical resolution at the 2 km height is about 260 m."
7) **Page 13, Line 19:** There was a mistake of section number. "…. (see Sec.22.2)." was changed on "… (see Sec.2.2.2)."
8) **Page 14, Line 6:** A needed space before the sentence "The occurrence of the liquid…" was added.
9) **Page 14, Line 7:** The name of parameter was corrected. "… cloud temperature" was replaced by "cloud top temperature".
10) We added the units for degree C on **Page 15, Line 3, Page 19, Lines 1-3**.
11)  **Page 16, Line 11:** "has larger occurrence values" was replaced by "has larger values of occurrence".
12) **Table 2:** The title of the table 2 "Cloudnet product used for this study" was replaced by "Cloudnet characteristics for Ny-Ålesund".

[revised manuscript text omitted]